# Byzantine-Robust Gossip: Insights from a Dual Approach

**Renaud Gaucher** *renaud.gaucher@polytechnique.edu*
*Center for Applied Mathematics*
*École polytechnique*
*IP Paris*
*Palaiseau*

**Aymeric Dieuleveut** *aymeric.dieuleveut@polytechnique.edu*
*Center for Applied Mathematics*
*École polytechnique*
*IP Paris*
*Palaiseau*

**Hadrien Hendrikx** *hadrien.hendrikx@inria.fr*
*Inria Center of Grenoble Alpes University*
*CNRS, LJK*
*Grenoble, France*

**Reviewed on OpenReview:** *https: // openreview. net/ forum? id= wrLiUpfk4s*

## Abstract

Distributed learning has many computational benefits but is vulnerable to attacks from a subset of devices transmitting incorrect information. This paper investigates Byzantine-resilient algorithms in a decentralized setting, where devices communicate directly in a peer-to-peer manner within a communication network. We leverage the so-called *dual approach* for decentralized optimization and propose a Byzantine-robust algorithm. We provide convergence guarantees in the average consensus subcase, discuss the potential of the dual approach beyond this subcase, and re-interpret existing algorithms using the dual framework. Lastly, we experimentally show the soundness of our method.

## 1 Introduction

Distributed optimization algorithms allow to harnessing the multiplication of devices to tackle the increasing computational complexity of machine learning models. Yet, the involvement of many parties in the learning process opens up the threat of misbehaving devices, that can purposely or inadvertently send arbitrarily harmful messages. Such device failures are commonly termed *Byzantine* (Lamport et al., 1982). This paper studies Byzantine robustness in the decentralized communication paradigm, which alleviates the risk of a single point of failure due to a central server by making devices directly communicate with each other in a peer-to-peer manner.

We consider devices (a.k.a nodes) that communicate synchronously within a communication network, abstracted as a graph $\mathcal{G} = (\mathcal{V}, \mathcal{E})$, where each vertex is a node that communicates with its neighbors in the graph through the edges. We denote as $\mathcal{V}_h$ the Byzantine nodes, and $\mathcal{V}_h = \mathcal{V} \backslash \mathcal{V}_b$ the remaining nodes, called honest. Similarly, we split the edges in $\mathcal{E} = \mathcal{E}_h \cup \mathcal{E}_b$, where $\mathcal{E}_h$ are the edges linking honest nodes, while edges in $\mathcal{E}_b$ link honest to Byzantine nodes. The subgraph of honest nodes $\mathcal{G}_h := (\mathcal{V}_h, \mathcal{E}_h)$ is always assumed to connected. For any $i \in \mathcal{V}_h$, we denote $f_i : \mathbb{R}^d \to \mathbb{R}$ the local objective function on node $i$, and we study the optimization problem:

$$\min_{x \in \mathbb{R}^d} \sum_{i \in \mathcal{V}_h} f_i(x). \tag{1}$$

We study decentralized algorithms to solve this problem, which rely on the *gossip* communication protocol (Boyd et al., 2006; Nedic & Ozdaglar, 2009; Duchi et al., 2011; Scaman et al., 2017), in which each node $i \in \mathcal{V}$ maintains a *local parameter* $x_i \in \mathbb{R}^d$, and updates this parameter by performing approximate average of this parameter with those of its neighbors in the graph.

Such gossip algorithms naturally arise when solving a well-chosen dual formulation of Equation (1) with standard first-order methods. This *dual approach* gives a principled framework to design and analyze efficient decentralized algorithms, leading for instance to acceleration (Scaman et al., 2017; Kovalev et al., 2020), variance reduction (Hendrikx et al., 2019) and has strong links with gradient tracking (Jakovetić, 2018). This paper investigates the benefits of this approach in the Byzantine robust setting.

**Related works.** Since its introduction by Blanchard et al. (2017), the setting of Byzantine-robust learning has been extensively studied. The large majority of works focus on the *federated* case, in which a central server organizes the training and communicates directly with each node (Yin et al., 2018; Chen et al., 2017; Alistarh et al., 2018; Li et al., 2019; El-Mhamdi et al., 2020; Karimireddy et al., 2023; 2021; Farhadkhani et al., 2022; Allouah et al., 2023). Some work investigated the robust decentralized optimization setting, either with fully connected networks (El-Mhamdi et al., 2021; Farhadkhani et al., 2023) or with more generic networks (He et al., 2023; Wu et al., 2023; Gaucher et al., 2025). However, those works only skimmed over the challenges of decentralized optimization in the Byzantine setting, and many key optimization techniques such as gradient tracking or acceleration are still unexplored in the presence of adversaries. Developing the dual framework in the presence of adversaries would thus open the way to tackle these challenges.

**Contributions.** In this paper, we (i) leverage the dual approach to derive EdgeClippedGossip, a decentralized robust algorithm based on clipped dual gradient descent, (ii) derive clipping rules and their associated convergence guarantees in the average consensus subproblem, in which EdgeClippedGossip recovers existing algorithms. Finally, (iii) we highlight the dynamics of the developed method compared to previous approaches and discuss its generalization beyond the average consensus case.

**Notations.** We denote $\mathbf{1}_n = (1,\ldots,1)^T$; $\|\cdot\|_2$ or simply $\|\cdot\|$ (resp. $\langle\cdot,\cdot\rangle$) the Euclidean norm (resp. inner product) on $\mathbb{R}^d$; $\|\cdot\|_F$ the Frobenius norm of any matrix in $\mathbb{R}^{d\times n}$, and $\langle M, N\rangle_F = \mathrm{Tr}(M^T N)$ the corresponding inner product. We denote $X_{i,:}$ the $i$-th row of a matrix $X$. For a matrix $M$ in $\mathbb{R}^{n\times d}$, we denote $\overline{M} := n^{-1}\mathbf{1}_n\mathbf{1}_n^T M$, and for $p \in \{1, 2, \infty\}$, we denote $\|M\|_{p,2}$, the $p$-norm of the vector of 2-norms of the rows of $M$, i.e., $\|(\|M_{i,:}\|_2)_{i\leqslant n}\|_p$. Moreover, $C^\dagger$ is the Moore Penrose inverse of $C$ and Id is the identity matrix. We identify $\mathbb{R}^{\mathcal{E}}$ to $\mathbb{R}^{|\mathcal{E}|}$ and $\mathbb{R}^{\mathcal{V}}$ to $\mathbb{R}^n$. We denote $a \wedge b = \min(a, b)$.

## 2 Deriving EdgeClippedGossip

This section describes a dual approach to Byzantine robustness. We first introduce a dual gossip algorithm, then show how it can be made robust using EdgeClippedGossip.

**Setting:** We denote $n_h = |\mathcal{V}_h|$ and $n_b = |\mathcal{V}_b|$ the total number of honest and Byzantine nodes, and $N_h(i)$ and $N_b(i)$ the number of honest and Byzantine neighbors of node $i$. Each agent owns and iteratively updates a local parameter $x_i^t$. At time $t$, we denote $X^t := [x_1^t, \ldots, x_n^t]^T \in \mathbb{R}^{\mathcal{V}\times d}$ the parameter matrix, which we split into honest and Byzantine as $X^t = \begin{pmatrix} X_h^t \\ X_b^t \end{pmatrix}$, where $X_h^t = [(x_i^t)_{i\in\mathcal{V}_h}]^T$ (resp. $X_b^t$) is the sub-matrix containing the models held by the honest workers (resp. Byzantine). Note that Byzantine nodes can declare different values to each of their neighbors. As such, only the number and positions of edges with Byzantine nodes matter, and we can w.l.o.g identify the number of edges linking Byzantine and honest nodes $|\mathcal{E}_b|$ to the number of Byzantine workers $n_b$.

### 2.1 Gossip - average consensus case

Gossip communications (Boyd et al., 2006) consist in sharing information between neighbors through local averaging steps. The standard gossip protocol writes $X^{t+1} = WX^t$, where $W \in [0,1]^{n\times n}$ is a *gossip matrix*.

**Definition 1** (Gossip matrix). A matrix $W \in \mathbb{R}_+^{n\times n}$ is a *gossip matrix* for a graph $\mathcal{G} = (\mathcal{V}, \mathcal{E})$ if: (i) $W$ is symmetric and doubly-stochastic: $W\mathbf{1}_n = \mathbf{1}_n$, and $\mathbf{1}_n^T W = \mathbf{1}_n^T$. (ii) $W$ is supported on the edges of the network: $W_{ij} \neq 0$ if and only if $i = j$ or $(i \sim j) \in \mathcal{E}$.

Such a matrix $W$ has eigenvalues $1 = \mu_1(W) \geqslant |\mu_2(W)| \geqslant \ldots \geqslant |\mu_n(W)|$. The propagation of information in the graph is characterized by the *spectral gap* of $W$, which is defined as $\gamma_W := 1 - |\mu_2(W)|$. Note that, if the graph $\mathcal{G}$ is connected, then $\gamma_W > 0$.

*Example* 1 (Gossip from Graph Laplacian). Consider the Laplacian matrix of the graph $L = D - A$ where $D$ is the diagonal matrix of degrees and $A$ is the adjacency matrix (i.e., $A_{ij} = 1_{(i \sim j) \in \mathcal{E}}$). Then, $W_L = \mathrm{Id} - \eta L$ is a gossip matrix for $\eta \leqslant 1/\mu_n(W)$, with spectral gap $\gamma_{W_L} = \eta \mu_{\min}^+(L)$, where $\mu_{\min}^+(L)$ is the second smallest eigenvalue of $L$.

The *average consensus* problem is an important special case of Equation (1).

**Definition 2.** The approximate **average consensus problem** (ACP) consists in computing $\overline{x^*} = \frac{1}{n_h} \sum_{i \in \mathcal{V}_h} x_i^*$, where each node $i \in \mathcal{V}_h$ holds a value $x_i^*$, which is equivalent to Problem (1) with $f_i(x) = \|x - x_i^*\|^2$. In the presence of Byzantines, only an approximate estimation of $\overline{x^*}$ can be found.

When there are no Byzantine nodes, standard gossip-averaging results ensure linear convergence of the parameters to the average (Boyd et al., 2006): $\sum_{i \in \mathcal{V}} \|x_i^t - \overline{x^*}\|^2 \leqslant (1 - \gamma)^{2t} \sum_{i \in \mathcal{V}} \|x_i^0 - \overline{x^*}\|^2$. Yet, this result is not robust to adversarial nodes: as the standard gossip communication uses averaging steps, it is *vulnerable to a single Byzantine node*, which can drive all honest nodes to any position (Blanchard et al., 2017).

## 2.2 Dual approach in decentralized optimization

The use of a dual formulation to solve decentralized optimization is very popular (Jakovetić et al., 2020; Uribe et al., 2020). Not only has it been at the heart of key advances such as acceleration (Scaman et al., 2017; Hendrikx et al., 2019; Kovalev et al., 2020) and variance reduction (Hendrikx et al., 2021), but it also allows us to understand methods such as gradient tracking (Jakovetić, 2018). It is thus natural to investigate whether the dual approach can be used for Byzantine robustness.

We first introduce the dual approach in a Byzantine-free setting. Then, we interpret dual variables and explain their use in the Byzantine-robust setting. Eventually, we instantiate the algorithm in the ACP setting and show that it recovers existing algorithms.

### 2.2.1 Byzantine-free dual approach.

We first show how the dual approach unrolls on Problem (1), in a setting without Byzantine agents. Let us write Problem (1) as a constrained optimization problem on $F(X) = \sum_{i \in \mathcal{V}} f_i(X_{i:})$, where the constraints write $X_{1:} = \ldots = X_{n:}$, or equivalently $\forall (i \sim j) \in \mathcal{E}, X_{i:} = X_{j:}$ for a connected graph. By encoding this constraint into a matrix $C^T \in \mathbb{R}^{\mathcal{E} \times \mathcal{V}}$, such that $[C^T X]_{(i \sim j),:} = X_{j:} - X_{i:}$ for any edge $(i \sim j)$ in $\mathcal{E}$, we obtain a *primal version* of Problem (1), which is, under convexity of functions $f_i$, equivalent to a *dual problem*:

$$\min_{X \in \mathbb{R}^{\mathcal{V} \times d}, \, C^T X = 0} F(X) = \min_{X \in \mathbb{R}^{\mathcal{V} \times d}} \max_{\Lambda \in \mathbb{R}^{\mathcal{E} \times d}} \left\{ F(X) - \langle \Lambda, C^T X \rangle \right\} \tag{Primal}$$

$$= - \min_{\Lambda \in \mathbb{R}^{\mathcal{E} \times d}} F^*(C\Lambda), \tag{Dual}$$

where $F^*(Y) := \sup_{X \in \mathbb{R}^{\mathcal{V} \times d}} \langle X, Y \rangle - F(X)$ is the Fenchel conjugate of $F$. Note that as $F$ is separable, $F^*$ can be decomposed as $F^*(Y) = \sum_{i \in \mathcal{V}} f_i^*(Y_{i:})$. The dual approach leverages Equation (Dual) to design decentralized optimization algorithms, for instance by solving the dual problem using gradient descent (GD). If we denote $Y := C\Lambda \in \mathbb{R}^{\mathcal{V} \times d}$ the dual variable deriving from the Lagrangian multipliers $\Lambda \in \mathbb{R}^{\mathcal{E} \times d}$, GD with step-size $\eta$ writes:

$$\Lambda^{t+1} = \Lambda^t - \eta C^T \nabla F^*(C\Lambda^t) \implies Y^{t+1} = Y^t - \eta C C^T \nabla F^*(Y^t). \tag{2}$$

**Dual gossip update.** The incidence matrix $C$ is a root of the Laplacian matrix $L$, as $C C^T = L$. It follows that in the case of the average consensus problem where $\nabla f_i^*(y) = y + x_i^*$, Equation (2) consists in doing an update on the dual variable $Y$, which for the associated primal variable $X := \nabla F^*(Y) = Y + X^*$ recovers the standard gossip protocol $X^{t+1} = (I - \eta L)X^t$.

**Convergence.** Using standard GD results, $\Lambda^t$ in Equation (2) converges linearly to $\Lambda^*$, the solution of (Dual), under strong convexity and smoothness assumptions on the $f_i$. It follows that $X^t := \nabla F^*(C\Lambda^t)$ converges to the solution of (Primal). Indeed:

---

**Algorithm 1** Byzantine-Resilient Decentralized Optimization with EdgeClippedGossip ($\boldsymbol{\tau}^t$)

---

**Input:** $\{f_j\}$, $\{y_j^0\}$, $\eta$, $\{\boldsymbol{\tau}^t\}$
**for** $t = 0$ **to** $T$ **do**
   **for** $j = 1, \ldots, n$ **in parallel do**
      Compute $x_j^t = \nabla f_j^*(y_j^t)$ if $j \in \mathcal{V}_h$ else $*$
      Share parameter $x_j^t$ with neighbors
      $y_j^{t+1} = y_j^t - \eta \sum_{i \sim j} \text{Clip}\left(x_j^t - x_i^t; \boldsymbol{\tau}_{(i \sim j)}^t\right)$

---

1. *Invariant.* As $Y^t \triangleq C\Lambda^t$, $\ker C^T = \text{span } \mathbf{1}$, and $\nabla F(X^t) = Y^t$, the sum of (primal) gradients is null by design:

$$\sum_{i \in \mathcal{V}_h} \nabla f_i(X_{i:}^t) = \mathbf{1}^T \nabla F(X^t) = \mathbf{1}^T Y^t = 0. \tag{Invariant}$$

2. *Asymptotic consensus.* Reaching a stationary point of Equation (2) means that $\nabla F^*(Y) \in \ker C^T$, thus $X_{i:} = X_{j:}$ for all $(i \sim j) \in \mathcal{E}$, i.e. consensus is reached among nodes.

Note that although dual gradients are arguably hard to compute in the general case, several approaches allow to leverage the (primal-)dual approach while either bypassing the problem or alleviating this cost (Hendrikx et al., 2020; Kovalev et al., 2020).

### 2.2.2 Duality in the Byzantine setting.

In (Dual), the Lagrangian multipliers $\Lambda^t \in \mathbb{R}^{\mathcal{E} \times d}$ correspond to the *influence* between nodes of the graph. Precisely, each row of $\Lambda$, indexed by edges as $\Lambda_{(i \sim j)}$, corresponds to the accumulated influence between nodes $i$ and $j$ through edge $(i \sim j)$. Indeed, the (primal) gradient is equal to the aggregation of neighboring Lagrangian multipliers, $\nabla f_i(x_i^t) \triangleq y_i^t = \sum_{(i \sim j)} \Lambda_{(i \sim j)}^t$, while the update of each $\Lambda_{(i \sim j)}^t$ multiplier involves only the two nodes of the edge: $\Lambda_{(i \sim j)}^{t+1} = \Lambda_{(i \sim j)}^t - \eta[\nabla f_j^*(y_j^t) - \nabla f_i^*(y_i^t)]$. This latter update thus corresponds to the *update of the influence* between node $i$ and node $j$ during a time step.

It follows that applying an operator on the update on the row $(i \sim j)$ of $\Lambda^t$ corresponds to altering the influence between nodes $i$ and $j$. For instance, setting the row $\Lambda_{(i \sim j)}$ to 0 corresponds to removing edge $(i \sim j)$ from the graph.

To address the vulnerability of plain gossip averaging to Byzantine parties, we regulate the influence update between nodes by clipping the update on the edges. Formally, for $\tau \geqslant 0$ and $x \in \mathbb{R}^d$, the projection of $x$ onto the $L_2$ ball of radius $\tau$ writes:

$$\text{Clip}(x; \tau) := \frac{x}{\|x\|_2} \min(\|x\|_2, \tau). \tag{3}$$

We thus consider for a general vector $\boldsymbol{\tau} \in \mathbb{R}^{\mathcal{E}}$, the regulation of the influence update on $(i \sim j)$, by setting $\Lambda_{(i \sim j)}^{t+1} = \Lambda_{(i \sim j)}^t - \eta \text{Clip}(\nabla f_j^*(y_j^t) - \nabla f_i^*(y_i^t), \boldsymbol{\tau}_{(i \sim j)}^t)$, which writes under matrix form as

$$\Lambda^{t+1} = \Lambda^t - \eta \text{Clip}\left(C^T \nabla F^*(C\Lambda^t), \boldsymbol{\tau}^t\right) \tag{4}$$

This gives Algorithm 1, called EdgeClippedGossip, which translates the update described above in terms of primal and dual iterates.

**Oracle strategy.** If the set of Byzantine edges $\mathcal{E}_b$ is known, then Equation (4) with $\tau_{(i \sim j)} = \infty \cdot 1_{(i \sim j) \notin \mathcal{E}_b}$ exactly recovers the generalized gossip algorithm 2 on the subgraph of honest nodes.

The previous derivations are summarized into the following result.

**Proposition 1.** *Consider a threshold $\tau > 0$ and $\boldsymbol{\tau} = \tau \boldsymbol{1}_{\mathcal{E}}$. Then, Algorithm 1 corresponds to a clipped gradient descent on the dual problem (Dual): $\Lambda^{t+1} = \Lambda^t - \eta \Pi_\tau \left(C^T \nabla F^*(C\Lambda^t)\right)$, where $\Pi_\tau$ is the projection on a ball of radius $\tau$ for the infinite-2 norm $\|M\|_{\infty,2} := \sup_{(i \sim j) \in \mathcal{E}} \|M_{(i \sim j)}\|_2$.*

Next, we instantiate the algorithm in the ACP case, in which the $\nabla f_i^*$ have a special form, and which is the setup under which theoretical guarantees are given in Sections 3 and 4.

### 2.2.3 EdgeClippedGossip for the ACP.

From now on, we focus on Algorithm 1 for the Average Consensus Problem (Definition 2).

We denote $X^* = [x_1^*, \ldots, x_n^*]^T$, the matrix of node-wise optimal parameters. In the ACP setting, $X^t = \nabla F^*(C\Lambda^t) \overset{\text{ACP}}{=} C\Lambda^t + X^*$, thus Equation (4) writes

$$\Lambda^{t+1} = \Lambda^t - \eta \operatorname{Clip}\left(C^T \nabla F^*(C\Lambda^t); \boldsymbol{\tau}^t\right) \Leftrightarrow X^{t+1} = X^t - \eta C \operatorname{Clip}(C^T X^t; \boldsymbol{\tau}^t). \tag{5}$$

In order to distinguish the impact of honest and Byzantine nodes, we decompose Matrix $C$ into blocks as $C = \begin{pmatrix} C_h & C_b \\ 0 & -\operatorname{Id}_{\mathcal{E}_b} \end{pmatrix}$, and $X$ as $\begin{pmatrix} X_h \\ X_b \end{pmatrix}$. The update on $(\Lambda^t)$ in Proposition 1 thus writes as

$$X_h^{t+1} = X_h^t - \eta(C_h\, C_b) \operatorname{Clip}(C^T X^t; \boldsymbol{\tau}^t); \quad X_b^{t+1} = *. \tag{6}$$

We analyze this sub-problem in the following section.

## 3 Global Clipping

We now investigate average consensus EdgeClippedGossip under a *global* clipping threshold, and highlight a choice of thresholds $(\boldsymbol{\tau}^t)_{t\geqslant 0}$ for which we derive convergence guarantees. All proofs for this section can be found in Appendix B.2.

### 3.1 The Global Clipping Rule

To analyze Equation (6), we decompose the influence update between honest and Byzantine edges as $(C^T X^t)_h = C_h^T X_h^t \in \mathbb{R}^{\mathcal{E}_h \times d}$ and $(C^T X^t)_b = C_b^T X_h^t - X_b^t \in \mathbb{R}^{\mathcal{E}_b \times d}$. For a given threshold $\tau^t$ we define as $\kappa^t$ the number of honest edges that are modified by $\operatorname{Clip}(\,\cdot\,, \tau^t)$, i.e.,

$$\kappa^t = \operatorname{card}\left\{(i \sim j) \in \mathcal{E}_h,\ \|(C_h^T X_h^t)_{(i\sim j)}\|_2 > \tau^t\right\}. \tag{7}$$

We then denote $\|C_h^T X_h^t\|_{1,2;\kappa^t}$ the $1, 2$-norm of the sub-matrix of clipped messages, i.e.:

$$\|C_h^T X_h^t\|_{1,2;\kappa^t} := \sum_{(i\sim j)\in\mathcal{E}_h,\ \kappa^t \text{ largest}} \|(C_h^T X_h^t)_{(i\sim j)}\|_2.$$

These quantities thus represent the fractions of weights of messages in $(C^T X)_h = ((x_i - x_j)_{(i\sim j)\in\mathcal{E}_h})$ that are affected by clipping. We now introduce

$$\Delta_\infty := \sup_{U_b \in \mathbb{R}^{\mathcal{E}_b \times d},\ \|U_b\|_{\infty,2}\leqslant 1} \|C_h^\dagger C_b U_b\|_{\infty,2},$$

which quantifies how much the Byzantine nodes can increase the variance of the honest nodes. We will need $\Delta_\infty < 1$ to obtain non-vacuous results, as otherwise, it means that Byzantine nodes introduce more variance than what is reduced by the contraction obtained through gossip averaging. $\Delta_\infty$ decreases when removing Byzantine edges from the graph, and can be bounded for some graphs.

**Lemma 1.** *For $\mathcal{G}_h$ fully connected where each honest node has exactly $N_b(i) = N_b$ Byzantine neighbors, $\Delta_\infty \leqslant \frac{2N_b}{n_h}\sqrt{d \wedge n_h N_b}$.*

To satisfy $\Delta_\infty < 1$, two regimes appear: it is sufficient that $n_h > 2N_b\sqrt{d}$ for small dimension $d$, but $n_h > 4N_b^3$ for large $d$, which is quickly untractable. Therefore, the limit fraction of Byzantine neighbors is significantly impacted by the dimension. We now define the following clipping threshold rule.

**Definition 3** (Global Clipping Rule (GCR)). Given a step size $\eta$, a sequence of clipping thresholds $(\tau^s)_{s \geqslant 0}$, satisfies the *Global Clipping Rule* if for all $t \geqslant 0$, either $\tau^t = 0$, or

$$\|C_h^T X_h^t\|_{1,2;\kappa^t} \geqslant \Delta_\infty \|C_h^T X_h^t\|_{1,2} + \eta \frac{|\mathcal{E}_b|^2}{n_h} \sum_{0 \leqslant s \leqslant t} \tau^s.$$

This clipping rule enforces that the clipping threshold is *below a certain value*, since decreasing $\tau^t$ increases $\kappa^t$, which in turn increases the left term. In particular, $\tau^t$ is reduced until the sum of the norms of *clipped* honest edges is greater than the sum of the norms of all honest edges, shrunk by the contraction $\Delta_\infty$, plus a term proportional to the sum of previous thresholds (which is known by all nodes). This latter term, $\eta |\mathcal{E}_b|^2 \sum_{s \leqslant t} \tau^s / n_h$ can be understood as a bound on the maximum *bias* induced by Byzantine nodes after $t$ steps, i.e. $1^T(\overline{X}_h^t - \overline{X}_h^0)$.

**Over-clipping.** The GCR defines an *upper bound* on $\tau^t$, but any value below this threshold works. By denoting $\kappa^{t,*}$ the smallest $\kappa^t$ such that the GCR holds, we must ensure that *at least* $\kappa^{t,*}$ honest edges are clipped. A solution to this end consists in clipping $\kappa^{t,*} + |\mathcal{E}_b|$ edges overall, by removing first the $|\mathcal{E}_b|$ largest edges, and computing $\kappa^{t,*}$ on the remaining edges afterwards. Note that the GCR still holds if the number of Byzantine edges is over-estimated. We provide in Appendix B.3 a simplified (yet looser) version of the GCR.

**Global coordination.** Implementing a global clipping threshold that satisfies Definition 3 requires some global coordination, which is hard in general in this decentralized setting. As such, we propose a proof of concept on what can be achieved by this approach under a favorable assumption of their communications, more than a practical decentralized algorithm.

## 3.2 Convergence result

Recall that $\mu_{\max}(L_h)$ is the maximal eigenvalue of the honest subgraph Laplacian, and $\Delta_\infty$ depends only on the topology of the full graph $\mathcal{G}$. We note $\overline{X}_h^* = \mathbf{1}_h \mathbf{1}_h^T X_h^* / n_h$

**Theorem 1.** *Assuming $\Delta_\infty < 1$, let $\eta$ be a constant step size: $\eta \leqslant 1/(1+|\mathcal{E}_h|(1-\Delta_\infty))\mu_{\max}(L_h)$. If the clipping thresholds $(\tau^s)_{s \geqslant 0}$ satisfy Definition 3 (GCR), then:*

$$\|X_h^{t+1} - \overline{X}_h^*\|_F^2 \leqslant \|X_h^t - \overline{X}_h^*\|_F^2 - \eta \sum_{i \sim j \in \mathcal{E}_h} \|x_i^t - x_j^t\|_2^2 \mathbf{1}_{\|x_i^t - x_j^t\|_2^2 \leqslant \tau^t}.$$

Theorem 1 ensures that the squared distance of honest models $X_h^t$ to the global optimum $\overline{X}_h^*$ decreases at each step, although not necessarily all the way to 0. In essence, what happens is the following: each step of EdgeClippedGossip pulls nodes closer together but at the cost of allowing Byzantine nodes to introduce some bias so that $\overline{X}_h^t - \overline{X}_h^* \neq 0$. This trade-off appears in the GCR: at some point, further reducing the variance is not beneficial because it introduces too much bias, and so the GCR enforces that $\tau^t \approx 0$, which essentially stops the algorithm. Yet, regardless of the initial heterogeneity, nodes benefit from running EdgeClippedGossip.

**Asymptotic error.** This theorem ensures that finding small $\tau^t$ that satisfy the GCR reduces the distance to the optimum, yet it does not characterize the asymptotic error. Furthermore, the smallest $\tau^t$ satisfying the GCR is an oracle quantity, in the sense that it requires computing a sum of node-wise differences on honest neighbors only.

# 4 Beyond global coordination and averaging

In this section, we investigate how to extend the dual approach beyond the average consensus case. We discuss de discrepancies between the developed dual framework and local clipping methods such as (He et al., 2023; Gaucher et al., 2025). Then we discuss on the difficulties to extend the dual framework beyond the average consensus setting.

### 4.1 Dual interpretation of existing algorithms

Several works investigated the problem of robust average consensus. Among those, He et al. (2023); Farhadkhani et al. (2023); Gaucher et al. (2025) propose robust communication algorithms that rely, under the hood, on performing gossip communication while altering the update on the Lagrangian multipliers at each step. All those algorithms can be written as:

$$\begin{cases} x_i^{t+1} = x_i^t - \eta F\big((x_i^t - x_j^t)_{j \sim i}\big) & \text{if } i \in \mathcal{V}_h \\ x_i^{t+1} = * & \text{if } i \in \mathcal{V}_b. \end{cases} \tag{F-RG}$$

where $F$ is an aggregation function that verifies a $(b, \rho)$-*robust summand properties*. The SOTA $(b, \rho)$-robust summand, Clipped Sum, relies on a clipping strategy: $\text{CS}(z_1, \ldots, z_n) := \sum_{i=1}^n \text{Clip}(z_i; \tau)$. Interestingly, the success of those methods relies partially on the control of the quantities $(x_i - x_j)_{i \sim j}$. In other words, they consist in altering the Lagrangian multipliers update $([C^T X^t]_{i \sim j})$ based on the norm of each entry.

**Proposition 2.** *When $\tau$ is a fixed constant,* (6) *is equivalent to Equation* (F-RG) *with Clipped Sum aggregation.*

**Local Clipping thresholds.** An other key element of the robustness of $\text{CS}_+$-RG, is that the clipping threshold is tailored locally, in a node-wise manner. Typically, the clipping threshold $\tau_i$ on a node $i$ is defined as the $2b$ largest value within $(\|x_i - x_j\|)_{j \sim i}$. It follows that the influence update of two neighbors may not be symmetric when $\text{Clip}(x_i - x_j; \tau_i) \neq \text{Clip}(x_i - x_j; \tau_j)$. This asymmetry between clipping thresholds in CS-RG may bias the average of honest parameters independently of the attack of Byzantine nodes.

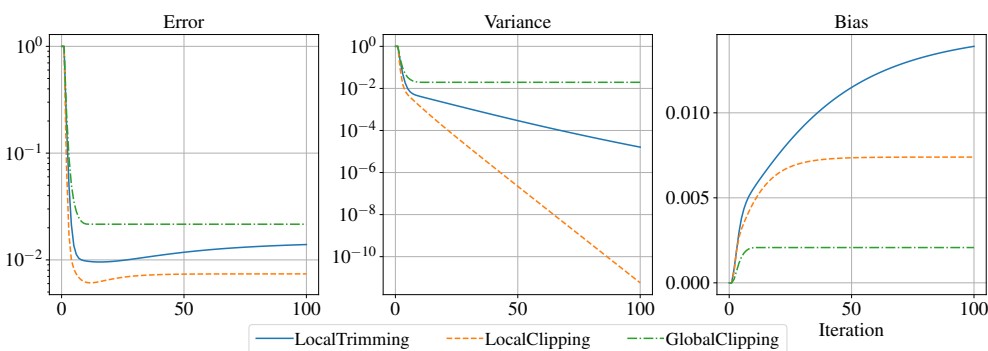

Figure 1: Bias - Variance decomposition of the parameters of honest nodes. GTS-RG (LocalTrimming), CS-RG (LocalClipping) and the GCR are tested under ALIE attack, in a Two Worlds graph (cf. Section 5) with 64 honest nodes in two cliques, each being neighbor to 16 nodes in the other clique, and to 3 Byzantine nodes.

**Trading bias and consensus.** This bias induced by the communication scheme is a fundamental difference with the dual approach such as exposed in Algorithm 1, where the bias is only induced by the Byzantines nodes. This discrepancy is reflected in the analysis: in the bias of Algorithm 1 is controlled directly in the GCR, while for instance (Gaucher et al., 2025) controls the additional bias at each step, and sum them up by using that honest nodes converges quickly enough to the same value. Overall, Algorithm 1 with the GCR does not alter the average of honest parameters and thus can induce less bias in the communication process than local clipping methods. Yet, there is no free lunch: the variance of the honest parameters does not converge to 0 using the GCR, while CS-RG for instance ensures asymptotic consensus. We illustrate this point in Figure 1.

### 4.2 Dual formalism for asymmetric clipping

Considering the weak convergence guarantee of the GCR, we modify slightly the dual formalism to allow local clipping thresholds. Then we discuss the major difficulties encountered when extending the dual approach beyond the case of average consensus.

**From symmetric to asymmetric clipping.** In Byzantine-free settings (see Section 2.2.1), dual gradient descent leads to gossip algorithms for which the influence of node $i$ on node $j$ is the opposite of the one of node $j$ on node $i$, both are stored in $\Lambda_{i\sim j}^t$. The matrix of influence updates $C^T\nabla F^*(C\Lambda^t)$ thus has one row for each edge, the update can be written as (2), and the (Invariant) property is kept. This symmetry between the influence of two honest nodes $(i,j) \in \mathcal{V}_h$ remains when the influence update is clipped, as long as both honest neighbors clip the influence update on edge $(i \sim j)$ with the same threshold. Allowing local clipping thresholds requires storing in two different variables the influence from node $i$ on node $j$ and from node $i$ to $j$.

To do so, we consider $\tilde{\mathcal{G}} = (\mathcal{V}, \tilde{\mathcal{E}})$ the directed version of the undirected graph $\mathcal{G}$, we enumerate its set of edges as $\tilde{\mathcal{E}} = \{i \to j\}$, and consider the *directed* incidence matrix $\tilde{C}^T \in \mathbb{R}^{\tilde{\mathcal{E}} \times \mathcal{V}}$, such that for $X \in \mathbb{R}^{\mathcal{V} \times d}$, and $i \to j \in \tilde{\mathcal{E}}$, $(\tilde{C}^T X)_{i\to j} = X_{j,:} - X_{i,:}$. Furthermore, we define $\tilde{B} \in \mathbb{R}^{\mathcal{V} \times \tilde{\mathcal{E}}}$ as the positive coordinates of the incidence matrix $\tilde{C}$, such that for $\tilde{\Lambda} \in \mathbb{R}^{\tilde{\mathcal{E}} \times d}$ and $j \in \mathcal{V}$, we have $[\tilde{B}\tilde{\Lambda}]_j = \sum_{(i\to j) \in \tilde{\mathcal{E}}} \tilde{\Lambda}_{i\to j}$. Under these notations, Equation (2) can be written with directed edge-wise clipping thresholds $\boldsymbol{\tau} \in \mathbb{R}^{\tilde{\mathcal{E}}}$ as:

$$\tilde{\Lambda}^{t+1} = \tilde{\Lambda}^t - \eta\,\mathrm{Clip}(\tilde{C}^T\nabla F^*(\tilde{B}\tilde{\Lambda}^t), \boldsymbol{\tau}^t) \implies \tilde{Y}^{t+1} = \tilde{Y}^t - \eta\tilde{B}\,\mathrm{Clip}(\tilde{C}^T\nabla F^*(\tilde{Y}^t); \boldsymbol{\tau}^t). \tag{8}$$

And, in the average consensus sub-case, Equation (5) generalizes as

$$\tilde{X}^{t+1} = \tilde{X}^t - \eta\tilde{B}\,\mathrm{Clip}(\tilde{C}^T\tilde{X}^t; \boldsymbol{\tau}^t). \tag{9}$$

**Node-wise clipping thresholds.** This formalism allows node-wise clipping thresholds as a sub-case of directed edge-wise clipping thresholds. Thus, communication algorithms with node-wise clipping thresholds as ClippedGossip in (He et al., 2023) or CS-RG in Gaucher et al. (2025) exactly correspond to Equation (9) for appropriate $\boldsymbol{\tau}^t$.

**Proposition 3.** *Denoting $M := \tilde{B}^\dagger \tilde{C}$, Equation (8) can be seen as a* clipped pre-conditioned *gradient descent on the dual objective $\Lambda \to F^*(C\Lambda)$:*

$$\tilde{\Lambda}^{t+1} = \tilde{\Lambda}^t - \eta\,\mathrm{Clip}(M^T\tilde{B}^T\nabla F^*(\tilde{B}\tilde{\Lambda}^t), \boldsymbol{\tau}). \tag{10}$$

*In particular, we expect fixed points of this iteration to be minimizers of $\tilde{\Lambda} \mapsto F^*(\tilde{B}\tilde{\Lambda})$ in the absence of Byzantines.*

**Loss of the invariant.** As denoted before, even in the absence of Byzantines, the (Invariant) property, i.e. $\mathbf{1}^T\tilde{Y}^t = 0$, does not hold *by design* anymore, since $\mathbf{1}^T\tilde{B} \neq 0$.

**Error metric.** The main difficulty in extending the dual approach to more generic loss functions, even with a global clipping threshold, lies in this loss of invariant. The Fenchel transform $F^*(Y_h^t)$ is no longer informative of the distance between $\nabla F^*(Y_h^t)$ and the solution of the problem (Primal) when $\mathbf{1}^T Y_h^t \neq 0$. Local clipping analyses such as F-RG bypass this difficulty by directly proving a linear decrease for the *variance* of honest nodes.

**Lack of Lyapunov function** Seeing *Equation* (8) as noisy variation of *Equation* (2) leads to writing $\tilde{Y}^{t+1} = \tilde{Y}^t - \eta L\nabla F^*(\tilde{Y}^t) + \eta E^t$, where $E^t$ is the error induced by both Byzantine corruption and clipping of honest edges. Note that $E^t$ may lie in the kernel of $L$. If it is possible to use F-RG analysis to upper-bound the error as $\|E^t\|^2 \leqslant \rho b\|\sqrt{L}\nabla F^*(\tilde{Y}_h^t)\|^2$, it is much more involved to find a proper Lyapunov function to leverage this bound. One can consider $F^*(PY^t)$, with $P$ being the projection on span $1^\perp$. However, at some point it is necessary to control $P\left(\nabla F^*(\tilde{Y}_h^t) - \nabla F^*(P\tilde{Y}_h^t)\right)$, which leads to vacuous results beyond the average consensus case.

**Conclusion.** In summary, the strength of the dual approach generally lies in its structure, which usually allows for building Lyapunov functions in a principled manner. The Byzantine case breaks this structure, which significantly hardens the analysis.

## 5 Experiments

We perform all experiments on a 'Two Worlds' graph with 64 nodes, made of two cliques of 32 honest nodes. Each node is connected to 16 nodes in the other clique. On such a graph, the theoretical maximal number

of tolerated Byzantine numbers per honest node is 16. The actual number of Byzantine neighbors per honest node depends on the experiment. We initialize the parameter of each honest node using a $N(0, I_5)$ distribution, each experimental configuration is run with 5 different seeds, and results are averaged. We implement ALIE (Baruch et al., 2019), FOE (Xie et al., 2020), Dissensus (He et al., 2023) and Spectral Heterogeneity (Gaucher et al., 2025), more details are available in Appendix A.1. We compare 1) the Global Clipping rule, the threshold is the largest clipping threshold verifying the GCR, 2) Local Clipping denotes the algorithm CS-RG (Gaucher et al., 2025), 3) Local Trimming denotes GTS-RG, i.e. the implementation of NNA (Farhadkhani et al., 2022) in the case of sparse communication networks (Gaucher et al., 2025).

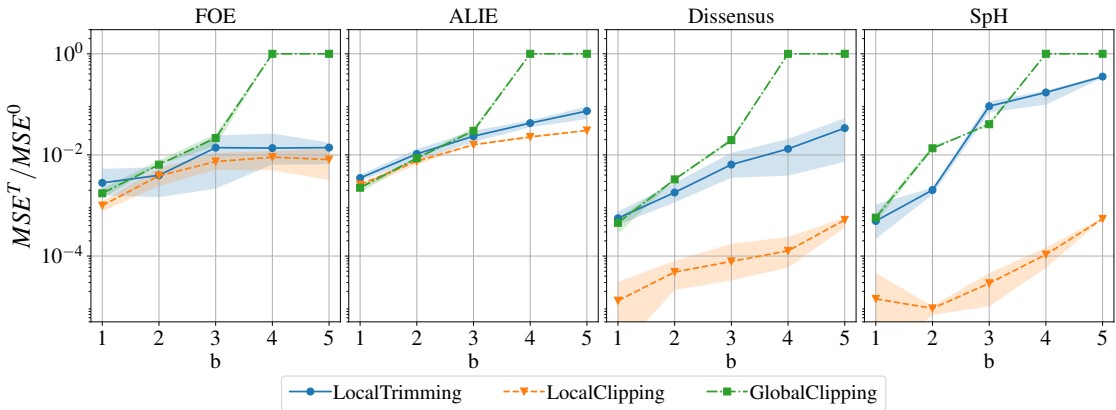

Figure 2: Relative mean square error after $T = 100$ communication steps, with a varying number of Byzantine neighbors to each honest node.

**Analysis.** We note on Figure 2 that the GCR has similar *worst-case*[1] performances on the investigated task than methods with local aggregation rules (i.e. GTS-RG and CS-RG). However, as soon as $\Delta_\infty \geqslant 1$, the GCR does not allow communication between nodes anymore, and we have $\mathrm{MSE}^T/\mathrm{MSE}^0 = 1$. This follows from the definition of the GCR itself: by design, the GCR is built such as to *ensure that nodes benefit from the communication.* As such, when there is no guarantee that this is the case anymore, the algorithms prevent communication between nodes. On the contrary, node-wise aggregation methods are provably robust up to a certain amount of Byzantines but do not have any guarantees beyond this breakdown point, and they do *increase* the mean square error by communicating – see e.g. Gaucher et al. (2025).

## 6 Conclusion

In this paper, we leverage the dual approach to design a byzantine-robust algorithm, which we then analyze in the average consensus case. We leverage the dual approach to re-interpret existing efficient robust communication algorithms, and we experimentally compare our algorithms with them. Finally, we highlight that the usual strength of the dual approach to decentralized learning - its structure - is broken by the introduction of Byzantine nodes, which brings significant challenges to the analysis. Yet, interesting algorithms arise, which do not ensure consensus among nodes (as standard algorithms do, at the price of some bias). Finding new ways of better characterizing the convergence properties of such algorithms is an interesting open problem.

## 7 Statement of Broader Impact

This paper presents work whose goal is to advance the field of Machine Learning. There are many potential societal consequences of our work, none which we feel must be specifically highlighted here.

---

[1]Note that the empirical worst-case performance among all attacks is the meaningful evaluation of the resilience of an algorithm, and thus the superior performance of LocalClipping against Dissensus and SpH only denotes that these attacks are not optimal against LocalClipping, and not that LocalClipping has superior robustness.

## Acknowledgments

The work of Aymeric Dieuleveut and Renaud Gaucher was supported by French State aid managed by the Agence Nationale de la Recherche (ANR) under France 2030 program with the reference ANR-23-PEIA-005 (REDEEM project). The work of Aymeric Dieuleveut was also supported by Hi!Paris - FLAG chair.

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

# A  Experimental design

## A.1  Byzantine attacks in the federated case

In our experiments, we implement SOTA attacks developed for the federated SGD setting: *Fall of empire* (FOE) from Xie et al. (2020) and *A little is enough* (ALIE) from Baruch et al. (2019), *Dissensus* from He et al. (2023) and *Spectral Heterogeneity* for Gaucher et al. (2025). We implement all these methods by making a Byzantine node $j \in \mathcal{V}_b$ declare to an honest neighbor $i \in \mathcal{V}_h$ a vector $x_j^t = x_i^t + \varepsilon_t^t a_i^t$, where $a_i^t \in \mathbb{R}^d$ is the attack direction on node $i$ and $\varepsilon_i^t \geqslant 0$ is the scale of the attack.

- **ALIE**. The Byzantine nodes compute the coordinate-wise standard deviation $\sigma^t$. Then they use $a_i^t = \sigma^t$.

- **FOE**. The Byzantine nodes compute the mean of the honest parameters $\overline{x^t}$, and declare $a_i^t = \overline{x^t}$.

- **Dissensus**. The Byzantines take $a_i^t = \sum_{k \sim j,\, k \in \mathcal{V}_h} x_i^t - x_k^t$.

- **Spectral Heterogeneity**. The Byzantines compute the Fiedler vector of the honest subgraph $e_{fied}$, i.e. the eigenvector associated with the second smallest eigenvalue of the Laplacian of the honest subgraph. Then, they take $a_i^t = [e_{fied}]_i \sum_{k \in \mathcal{V}_h} [e_{fied}]_k x_k^t$.

In all attacks we chose $\varepsilon_i^t$ to maximize the error induced by Byzantines: for trimming we take it such that all Byzantine attacks are just below the trimming threshold, while for clipping strategies we chose $e_i^t >> 1$.

## B Proofs

### B.1 Notations

In this subsection we resume the different notations of the proofs. Table 1 summarizes notations used for primal and dual parameters, as well as Lagrange multipliers.

Table 1: Summary of notations

| Variable | Name | Size | Rows |
|---|---|---|---|
| $X$ | primal parameter | $\mathcal{V} \times d$ | $(x_i)_{i \in \mathcal{V}}$ |
| $Y$ | dual parameter | $\mathcal{V} \times d$ | $(y_i)_{i \in \mathcal{V}}$ |
| $\Lambda$ | Lagrange multipliers | $\mathcal{E} \times d$ | $(\lambda_{(i \sim j)})_{(i \sim j) \in \mathcal{E}}$ |
| $C^T \nabla F^*(C\Lambda)$ | Influence update on the edges | $\mathcal{E} \times d$ | $(x_i - x_j)_{(i \sim j) \in \mathcal{E}}$ |

We recall the algorithm in the global clipping setting:

$$X_h^{t+1} = X_h^t - \eta(C_h\, C_b)\, \text{Clip}(C^T X^t; \boldsymbol{\tau}^t)$$
$$X_b^{t+1} = *.$$

In the global clipping setting, we consider that, as $\boldsymbol{\tau}^t = \tau^t \mathbf{1}_{\mathcal{E}}$, we can overload the clipping operators writing for $\Lambda_h \in \mathbb{R}^{\mathcal{E}_h \times d}$ and $\Lambda_b \in \mathbb{R}^{\mathcal{E}_b \times d}$,

$$\text{Clip}(\Lambda_h; \tau^t) = [\text{Clip}(\lambda_{(i \sim j)}, \tau^t)]_{(i \sim j) \in \mathcal{E}_h} \tag{11}$$
$$\text{Clip}(\Lambda_b; \tau^t) = [\text{Clip}(\lambda_{(i \sim j)}, \tau^t)]_{(i \sim j) \in \mathcal{E}_b} \tag{12}$$

We use the following notations:

- For any matrix $M \in \mathbb{R}^{\mathcal{V}_h \times d}$, we denote

$$\overline{M} := \frac{1}{n_h} \mathbf{1}_{n_h} \mathbf{1}_{n_h}^T M.$$

- We note the primal parameter re-centered on the solution

$$Z_h^t := X_h^t - \overline{X_h^*}.$$

- The unitary matrix of attack on edges linking to Byzantine nodes

$$U_b^t := \frac{1}{\eta \tau^t} (\Lambda_b^{t+1} - \Lambda_b^t).$$

Thus $\|U_b^t\|_{\infty,2} \leqslant 1$ and $\Lambda_b^{t+1} = \Lambda_b^t - \eta\, \text{Clip}([C^T X^t]_b; \tau^t) = \Lambda_b^t + \eta \tau^t U_b^t$.

- The error term induced by clipping and Byzantine nodes

$$E^t := C_h \left( C_h^T Z_h^t - \text{Clip}(C_h^T Z_h^t; \tau^t) \right) + \tau^t C_b U_b^t,$$

- We denote for $p \in \{1, 2\}$ and $\kappa^t \in \{0, \ldots, |\mathcal{E}_h|\}$

$$\|C_h^T X_h^t\|_{p;\kappa^t}^p := \sum_{(i \sim j) \in \mathcal{E}_h, \ \kappa^t \text{ largest}} \|(C_h^T X_h^t)_{(i \sim j)}\|_2^p$$

$$\|C_h^T X_h^t\|_{p;-\kappa^t}^p := \|C_h^T X_h^t\|_p^p - \|C_h^T X_h^t\|_{p;\kappa^t}^p.$$

- We recall that by default $\|\cdot\|$ and $\|\cdot\|_2$ is the usual Euclidean norm (or the Frobenius norm in case of matrix).

And we recall some previous facts:

- $L_h = C_h C_h^T = \frac{1}{2}\tilde{C}_h \tilde{C}_h^T$ the Lagrangian matrix on the honest subgraph $\mathcal{G}_h$.
- $C_h^T \mathbf{1}_{n_h} = 0 \implies C_h^T \overline{X_h^t} = 0 \implies C_h^T X_h^t = C_h^T Z_h^t$ .
- $C_h^\dagger$ is the Moore-Penrose pseudo inverse of $C_h$.

## B.2 Analysis of Global Clipping Rule (GCR) Theorem 1

**Lemma 2.** *(Gossip - Error decomposition) The iterate update can be decomposed as*

$$Z_h^{t+1} = (I_h - \eta L_h)Z_h^t + \eta E^t.$$

*Proof.* The actualization of $X_h^t$ writes

$$X_h^{t+1} = X_h^t - \eta C_h \text{Clip}(C_h^T X_h^t; \tau^t) - \eta C_b \text{Clip}([C^T X^t]_b; \tau^t).$$

Using $C_h^T X_h^t = C_h^T Z_h^t$, we get

$$\begin{aligned}
Z_h^{t+1} &= Z_h^t - \eta C_h \text{Clip}(C_h^T X_h^t; \tau^t) + \eta \tau^t C_b U_b^t \\
&= (I_h - \eta C_h C_h^T)Z_h^t + \eta \left\{ C_h \left( C_h^T Z_h^t - \text{Clip}(C_h^T Z_h^t; \tau^t) \right) + \tau^t C_b U_b^t \right\} \\
&= (I_h - \eta L_h)Z_h^t + \eta E^t.
\end{aligned}$$

$\square$

**Lemma 3.** *(Sufficient decrease on the variance) We have the decomposition of the variance as a biased gradient descent:*

$$\|Z_h^{t+1} - \overline{Z_h^t}\|^2 = \|Z_h^t - \overline{Z_h^t}\|^2 + \eta^2 \|L_h Z_h^t - E^t\|^2 - 2\eta\|C_h^T Z_h^t\|^2 + 2\eta\langle Z_h^t - \overline{Z_h^t}, E^t\rangle.$$

*Proof.* we start from Lemma 2

$$\begin{aligned}
\|Z_h^{t+1} - \overline{Z_h^t}\|^2 &= \|Z_h^t - \overline{Z_h^t} - \eta(L_h Z_h^t - E^t)\|^2 \\
&= \|Z_h^t - \overline{Z_h^t}\|^2 + \eta^2\|L_h Z_h^t - E^t\|^2 - 2\eta\langle L_h Z_h^t, Z_h^t - \overline{Z_h^t}\rangle + 2\eta\langle Z_h^t - \overline{Z_h^t}, E^t\rangle \\
&= \|Z_h^t - \overline{Z_h^t}\|^2 + \eta^2\|L_h Z_h^t - E^t\|^2 - 2\eta\|C_h^T Z_h^t\|^2 + 2\eta\langle Z_h^t - \overline{Z_h^t}, E^t\rangle,
\end{aligned}$$

where we used that $C_h^T \overline{Z_h^t} = 0$ and $C_h C_h^T = L_h$. $\square$

**Lemma 4.** *(complete sufficient decrease)*

$$\|Z_h^{t+1}\|^2 = \|Z_h^t\|^2 + \eta^2\|L_h Z_h^t - E^t\|^2 - 2\eta\|C_h^T Z_h^t\|^2 + 2\eta\langle Z_h^t - \overline{Z_h^t}, E^t\rangle + 2\langle \overline{Z_h^t}, \overline{Z_h^{t+1}} - \overline{Z_h^t}\rangle.$$

*Proof.* We leverage that

$$
\begin{aligned}
\|Z_h^{t+1}\|^2 &= \|Z_h^{t+1} - \overline{Z_h^t} + \overline{Z_h^t}\|^2 \\
&= \|Z_h^{t+1} - \overline{Z_h^t}\|^2 + \|\overline{Z_h^t}\|^2 + 2\langle \overline{Z_h^t}, Z_h^{t+1} - \overline{Z_h^t}\rangle \\
&= \|Z_h^{t+1} - \overline{Z_h^t}\|^2 + \|\overline{Z_h^t}\|^2 + 2\langle \overline{Z_h^t}, \overline{Z_h^{t+1}} - \overline{Z_h^t}\rangle.
\end{aligned}
$$

Then we use Lemma 3. $\qquad\square$

**Assumption 1.** Definition 3 Trade-off between clipping error and Byzantine influence. By denoting

$$
\Delta_\infty := \sup_{\substack{U_b \in \mathbb{R}^{\mathcal{E}_b \times d} \\ \|U_b\|_{\infty,2} \leqslant 1}} \|C_h^\dagger C_b U_b\|_{\infty,2},
$$

We assume that $\tau^t = 0$, or

$$
\|C_h^T X_h^t\|_{\kappa^t} \geqslant \Delta_\infty \|C_h^T X_h^t\|_1 + \eta \frac{|\mathcal{E}_b|^2}{n_h} \sum_{0 \leqslant s \leqslant t} \tau^s.
$$

**Lemma 5.** *(Join control of the first order error and second order bias)* *Under Definition 3, we have that*

$$
\eta\langle Z_h^t - \overline{Z_h^t}, E^t\rangle + \langle \overline{Z_h^t}, \overline{Z_h^{t+1}} - \overline{Z_h^t}\rangle + \eta^2\|\overline{E^t}\|_2^2 \leqslant \eta\|C_h^T Z_h^t\|_{\kappa^t}^2,
$$

*or* $\tau^t = 0$.

*Proof.* We begin by upper bounding precisely both error terms:

1. Variance-induced error.

$$
\begin{aligned}
\langle Z_h^t - \overline{Z_h^t}, E^t\rangle &= \langle Z_h^t - \overline{Z_h^t}, C_h\left(C_h^T Z_h^t - \mathrm{Clip}(C_h^T Z_h^t; \tau^t)\right)\rangle + \langle Z_h^t - \overline{Z_h^t}, \tau^t C_b U_b^t\rangle \\
&= \langle C_h^T Z_h^t, C_h^T Z_h^t - \mathrm{Clip}(C_h^T Z_h^t; \tau^t)\rangle + \langle C_h^T Z_h^t, \tau^t C_h^\dagger C_b U_b^t\rangle,
\end{aligned}
$$

Using that $C_h^T \overline{Z_h^t} = 0$. On one side we have that,

$$
\begin{aligned}
\langle C_h^T Z_h^t, C_h^T Z_h^t - \mathrm{Clip}(C_h^T Z_h^t; \tau^t)\rangle &= \sum_{(i\sim j)\in\mathcal{E}_h} \langle z_i^t - z_j^t, z_i^t - z_j^t - \mathrm{Clip}(z_i^t - z_j^t; \tau^t)\rangle \\
&= \sum_{(i\sim j)\in\mathcal{E}_h} \|z_i^t - z_j^t\|_2\left(\|z_i^t - z_j^t\|_2 - \tau^t\right)_+.
\end{aligned}
$$

One the other side, by using that, for $M, N \in \mathbb{R}^{\mathcal{E}_h \times d}$, $\langle M, N\rangle \leqslant \|M\|_{1,2}\|N\|_{\infty,2}$, as:

$$
\begin{aligned}
\langle M, N\rangle &= \sum_{(i\sim j)\in\mathcal{E}_h} \langle M_{(i\sim j)}, N_{(i\sim j)}\rangle \\
&\leqslant \sum_{(i\sim j)\in\mathcal{E}_h} \|M_{(i\sim j)}\|_2\|N_{(i\sim j)}\|_2 \\
&\leqslant \|N\|_{\infty,2} \sum_{(i\sim j)\in\mathcal{E}_h} \|M_{(i\sim j)}\|_2,
\end{aligned}
$$

we can upper bound the second term

$$
\langle C_h^T Z_h^t, \tau^t C_h^\dagger C_b U_b^t\rangle \leqslant \tau^t\|C_h^T Z_h^T\|_{1,2}\|C_h^\dagger C_b U_b^t\|_{\infty,2}.
$$

Then by defining $\Delta_\infty = \sup_{U_b \in \mathbb{R}^{\varepsilon_b \times d}} \|C_h^\dagger C_b U_b^t\|_{\infty,2}$, using $\|C_h^T Z_h^t\|_{1,2} = \sum_{(i \sim j) \in \mathcal{E}_h} \|z_i^t - z_j^t\|_2$, we have that

$$\langle Z_h^t - \overline{Z_h^t}, E^t \rangle \leqslant \sum_{(i \sim j) \in \mathcal{E}_h} \|z_i^t - z_j^t\|_2 \left( \|z_i^t - z_j^t\|_2 - \tau^t \right)_+ + \tau^t \Delta_\infty \sum_{(i \sim j) \in \mathcal{E}_h} \|z_i^t - z_j^t\|_2$$

$$= \sum_{(i \sim j) \in \mathcal{E}_h} \|z_i^t - z_j^t\|_2^2 \, \mathbf{1}_{\|z_i^t - z_j^t\|_2 > \tau^t}$$

$$- \tau^t \left( \sum_{(i \sim j) \in \mathcal{E}_h} \|z_i^t - z_j^t\|_2 \, \mathbf{1}_{\|z_i^t - z_j^t\|_2 > \tau^t} - \Delta_\infty \sum_{(i \sim j) \in \mathcal{E}_h} \|z_i^t - z_j^t\|_2^2 \right).$$

Let's consider any $\kappa^t \in \mathbb{N}$ such that

$$\kappa^t \in \left[ \sum_{(i \sim j) \in \mathcal{E}_h} \mathbf{1}_{\|z_i^t - z_j^t\|_2 > \tau^t}, \sum_{(i \sim j) \in \mathcal{E}_h} \mathbf{1}_{\|z_i^t - z_j^t\|_2 \geqslant \tau^t} - 1 \right].$$

Then, using

$$\|C_h^T Z_h^t\|_{p,\kappa^t}^p := \sum_{\substack{(i \sim j) \in \mathcal{E}_h \\ \kappa^t \text{ largest}}} \|z_i^t - z_j^t\|_2^p,$$

the previous inequality writes

$$\langle Z_h^t - \overline{Z_h^t}, E^t \rangle \leqslant \|C_h^T Z_h^t\|_{2,\kappa^t}^2 - \tau^t \left( \|C_h^T Z_h^t\|_{1,\kappa^t} - \Delta_\infty \|C_h^T Z_h^t\|_1 \right).$$

From this upper bound we could derive a clipping rule taking only into account the error induced by clipping and Byzantine nodes on the variance. In this analysis, we will include the second error term induced by the bias.

2. Bias-induced error.
We consider the term $\langle \overline{Z_h^t}, \overline{Z_h^{t+1}} - \overline{Z_h^t} \rangle$. We remark that, using that $Z_h^{t+1} = (I_h - C_h C_h^T) Z_h^t + \eta E^t$, and considering that $\mathbf{1}_{n_h}^T C_h = 0$, using the definition of $E^t$ we get:

$$\mathbf{1}_{n_h}^T Z_h^{t+1} = \mathbf{1}_{n_h}^T Z_h^t + \eta \tau^t \mathbf{1}_{n_h}^T C_b U_b^t = \sum_{s=0}^t \eta \tau^s \mathbf{1}_{n_h}^T C_b U_b^s.$$

Using this, we remark that

$$\eta^2 \|\overline{E^t}\|_2^2 = \|\overline{Z^{t+1}} - \overline{Z^t}\|_2^2 \implies \eta^2 \|\overline{E^t}\|_2^2 + \langle \overline{Z_h^t}, \overline{Z_h^{t+1}} - \overline{Z_h^t} \rangle = \langle \overline{Z_h^{t+1}}, \overline{Z_h^{t+1}} - \overline{Z_h^t} \rangle.$$

Thus we get

$$\langle \overline{Z_h^{t+1}}, \overline{Z_h^{t+1}} - \overline{Z_h^t} \rangle = \left\langle \sum_{s=0}^t \eta \tau^s \frac{1}{n_h} \mathbf{1}_{n_h} \mathbf{1}_{n_h}^T C_b U_b^s, \eta \tau^t \frac{1}{n_h} \mathbf{1}_{n_h} \mathbf{1}_{n_h}^T C_b U_b^t \right\rangle$$

$$= \frac{\tau^t \eta^2}{n_h} \sum_{s=0}^t \tau^s \left\langle \mathbf{1}_{n_h}^T C_b U_b^s, \mathbf{1}_{n_h}^T C_b U_b^t \right\rangle$$

$$= \frac{\tau^t \eta^2}{n_h} \sum_{s=0}^t \tau^s \left\langle \sum_{i \in \mathcal{V}_h} N_b(i) u_i^s, \sum_{i \in \mathcal{V}_h} N_b(i) u_i^t \right\rangle.$$

Where $u_i^t$ is the mean vector of Byzantine neighbors of node $i \in \mathcal{V}_h$, so that $\|u_i^t\|_2 \leqslant 1$, thus we get

$$\langle \overline{Z_h^t}, \overline{Z_h^{t+1}} - \overline{Z_h^t} \rangle \leqslant \frac{\tau^t \eta^2}{n_h} \sum_{s=0}^t \tau^s \left( \sum_{i \in \mathcal{V}_h} N_b(i) \right)^2 = \frac{\eta^2 \tau^t |\mathcal{E}_b|^2}{n_h} \sum_{s=0}^t \tau^s.$$

Now that we controlled both terms, we can mix both error terms:

$$\langle Z_h^t - \overline{Z_h^t}, E^t\rangle + \eta^{-1}\langle \overline{Z_h^t}, \overline{Z_h^{t+1}} - \overline{Z_h^t}\rangle$$

$$\leqslant \|C_h^T Z_h^t\|_{2,\kappa^t}^2 - \tau^t \left(\|C_h^T Z_h^t\|_{1,\kappa^t} - \Delta_\infty\|C_h^T Z_h^t\|_1\right) + \tau^t \frac{\eta|\mathcal{E}_b|^2}{n_h}\sum_{s=0}^{t-1}\tau^s$$

$$= \|C_h^T Z_h^t\|_{2,\kappa^t}^2 - \tau^t \left(\|C_h^T Z_h^t\|_{1,\kappa^t} - \Delta_\infty\|C_h^T Z_h^t\|_1 - \frac{\eta|\mathcal{E}_b|^2}{n_h}\sum_{s=0}^{t}\tau^s\right).$$

Hence, as long as

$$\|C_h^T Z_h^t\|_{1,\kappa^t} \geqslant \Delta_\infty\|C_h^T Z_h^t\|_1 + \frac{\eta|\mathcal{E}_b|^2}{n_h}\sum_{s=0}^{t}\tau^s, \tag{13}$$

we have that

$$\langle Z_h^t - \overline{Z_h^t}, E^t\rangle + \eta^{-1}\langle \overline{Z_h^{t+1}}, \overline{Z_h^{t+1}} - \overline{Z_h^t}\rangle \leqslant \|C_h^T Z_h^t\|_{2,\kappa^t}^2.$$

When Equation (13) cannot be enforced using $\kappa^t \leqslant |\mathcal{E}_h| - 1$, then we take $\tau^t = 0$ (thus $\kappa^t = |\mathcal{E}_h|$), everything is clipped and nodes parameters don't move anymore. □

**Lemma 6.** *(Control of first and 2nd order error terms together) Assume Definition 3 and that*

$$\eta \leqslant \frac{1}{\mu_{\max}(L_h)},$$

*then by denoting the first and 2nd order error term, divided by $\eta$ as*

$$\zeta^t := \eta\|L_h Z_h^t - E^t\|^2 + 2\langle Z_h^t - \overline{Z_h^t}, E^t\rangle + 2\eta^{-1}\langle \overline{Z_h^t}, \overline{Z_h^{t+1}} - \overline{Z_h^t}\rangle.$$

*We have the control:*

$$\zeta^t \leqslant \eta\mu_{\max}(L_h)\left(\|C_h^T Z_h^t\|_{2,-\kappa^t}^2 + (\tau^t)^2\left[\kappa^t(1 - 2\Delta_\infty) + \Delta_\infty^2|\mathcal{E}_h|\right]\right) + 2\|C_h^T Z_h^t\|_{2,\kappa^t}^2.$$

*Proof.* This part of the proof allows to compute the maximum stepsize usable. We want to control the discretization term $\eta^2\|L_h Z_h^t - E^t\|_2^2$. If $E^t = 0$ it would have been done using a step-size small enough. In this case we will do the same, but we will monitor the interactions between the discretization and the control of the error. Note that

$$E^t - L_h Z_h^t = C_h \operatorname{Clip}(C_h^T Z_h^t; \tau^t) + \tau^t C_b U_b^t, \tag{14}$$

and that at the first order $\langle C_g \operatorname{Clip}(C_g^T Z_h^t; \tau^t), \tau^t C_b U_b^t\rangle \leqslant 0$ if Byzantine aims at preventing the convergence. Thus, we will include this second-order term in the proof of the Lemma 3.

We denote by $\mu_{\max}(L_h)$ the largest eigenvalue of $L_h$.

We denote the first and 2nd order error term, divided by $\eta$ as

$$\zeta^t := \eta\|L_h Z_h^t - E^t\|^2 + 2\langle Z_h^t - \overline{Z_h^t}, E^t\rangle + 2\eta^{-1}\langle \overline{Z_h^t}, \overline{Z_h^{t+1}} - \overline{Z_h^t}\rangle.$$

Hence $\|Z_h^{t+1}\|^2 \leqslant \|Z_h^t\|^2 + \eta\zeta^t - 2\eta\|C_h^T Z_h^t\|^2.$

Using

$$\|E^t - L_h Z_h^t\| = \|C_h(\operatorname{Clip}(C_h^T Z_h^t; \tau^t) - \tau^t C_b^\dagger C_b U_b^t) + \overline{E^t}\|^2$$
$$= \|C_h(\operatorname{Clip}(C_h^T Z_h^t; \tau^t) - \tau^t C_b^\dagger C_b U_b^t)\|^2 + \|\overline{E^t}\|^2,$$

we have:

$$\zeta^t = \eta\|C_h(\mathrm{Clip}(C_h^T Z_h^t; \tau^t) - \tau^t C_h^\dagger C_b U_b^t)\|_2^2 + \eta\|\overline{E^t}\|_2^2$$
$$+ 2\langle Z_h^t - \overline{Z_h^t}, E^t\rangle + 2\eta^{-1}\langle \overline{Z_h^t}, \overline{Z_h^{t+1}} - \overline{Z_h^t}\rangle$$
$$\leqslant \eta\mu_{\max}(L_h)\left(\|\mathrm{Clip}(C_h^T Z_h^t; \tau^t)\|^2 + \|\tau^t C_h^\dagger C_b U_b^t\|^2\right)$$
$$- 2\eta\mu_{\max}(L_h)\langle \mathrm{Clip}(C_h^T Z_h^t) - C_h^T Z_h^t + C_h^T Z_h^t, \tau^t C_h^\dagger C_b U_b^t\rangle$$
$$+ 2\langle C_h^T Z_h^t, \tau^t C_h^\dagger C_b U_b^t\rangle + 2\langle C_h^T Z_h^t, C_h^T Z_h^t - \mathrm{Clip}(C_h^T Z_h^t; \tau^t)\rangle + 2\eta^{-1}\langle \overline{Z_h^{t+1}}, \overline{Z_h^{t+1}} - \overline{Z_h^t}\rangle$$
$$= \eta\mu_{\max}(L_h)\left(\|\mathrm{Clip}(C_h^T Z_h^t; \tau^t)\|^2 + \|\tau^t C_h^\dagger C_b U_b^t\|^2\right)$$
$$+ 2\langle C_h^T Z_h^t, C_h^T Z_h^t - \mathrm{Clip}(C_h^T Z_h^t; \tau^t)\rangle + 2\eta^{-1}\langle \overline{Z_h^{t+1}}, \overline{Z_h^{t+1}} - \overline{Z_h^t}\rangle$$
$$+ 2(1 - \eta\mu_{\max}(L_h))\underbrace{\langle C_h^T Z_h^t, \tau^t C_h^\dagger C_b U_b^t\rangle}_{\leqslant \|C_h^T Z_h^t\|_1 \Delta_\infty \tau^t}$$
$$+ 2\eta\mu_{\max}(L_h)\underbrace{\langle C_h^T Z_h^t - \mathrm{Clip}(C_h^T Z_h^t; \tau^t), \tau^t C_h^\dagger C_b U_b^t\rangle}_{\leqslant \|C_h^T Z_h^t - \mathrm{Clip}(C_h^T Z_h^t;\tau^t)\|_1 \Delta_\infty \tau^t}.$$

Then we use that
$$\|C_h^T Z_h^t - \mathrm{Clip}(C_h^T Z_h^t; \tau^t)\|_1 = \|C_h^T Z_h^t\|_1 - \|\mathrm{Clip}(C_h^T Z_h^t; \tau^t)\|_1,$$

to get, for $\eta \leqslant \mu_{\max}(L_h)$,

$$\zeta^t \leqslant \eta\mu_{\max}(L_h)\left(\|\mathrm{Clip}(C_h^T Z_h^t; \tau^t)\|^2 + \|\tau^t C_h^\dagger C_b U_b^t\|^2\right)$$
$$+ 2\langle C_h^T Z_h^t, C_h^T Z_h^t - \mathrm{Clip}(C_h^T Z_h^t; \tau^t)\rangle \quad + 2\eta^{-1}\langle \overline{Z_h^{t+1}}, \overline{Z_h^{t+1}} - \overline{Z_h^t}\rangle$$
$$+ 2(1 - \eta\mu_{\max}(L_h))\|C_h^T Z_h^t\|_1 \Delta_\infty \tau^t + 2\eta\mu_{\max}(L_h)\|C_h^T Z_h^t\|_1 \Delta_\infty \tau^t$$
$$- 2\eta\mu_{\max}(L_h)\|\mathrm{Clip}(C_h^T Z_h^t; \tau^t)\|_1 \Delta_\infty \tau^t$$
$$\leqslant \eta\mu_{\max}(L_h)\left(\|\mathrm{Clip}(C_h^T Z_h^t; \tau^t)\|^2 + \|\tau^t C_h^\dagger C_b U_b^t\|^2 - 2\|\mathrm{Clip}(C_h^T Z_h^t; \tau^t)\|_1 \Delta_\infty \tau^t\right)$$
$$+ 2\langle C_h^T Z_h^t, C_h^T Z_h^t - \mathrm{Clip}(C_h^T Z_h^t; \tau^t)\rangle + 2\|C_h^T Z_h^t\|_1 \Delta_\infty \tau^t + 2\eta^{-1}\langle \overline{Z_h^{t+1}}, \overline{Z_h^{t+1}} - \overline{Z_h^t}\rangle.$$

Then, the second term is controlled using the proof of Lemma 5.

By denoting
$$\Delta_2^2 := \sup_{\|U_b^t\|_{\infty,2}\leqslant 1} \|C_h^\dagger C_b U_b^t\|_2^2,$$

we have that, under Definition 3

$$\zeta^t \leqslant \eta\mu_{\max}(L_h)\left(\|C_h^T Z_h^t\|_{2,-\kappa^t}^2 + \tau^t\left(\tau^t\kappa^t + \tau^t\Delta_2^2 - 2\tau^t\kappa^t\Delta_\infty - 2\Delta_\infty\|C_h^T Z_h^t\|_{1,-\kappa^t}\right)\right) + 2\|C_h^T Z_h^t\|_{2,\kappa^t}^2$$
$$\leqslant \eta\mu_{\max}(L_h)\left((1 - \Delta_\infty)\|C_h^T Z_h^t\|_{2,-\kappa^t}^2 + (\tau^t)^2\left[\kappa^t(1 - 2\Delta_\infty) + \Delta_2^2\right]\right) + 2\|C_h^T Z_h^t\|_{2,\kappa^t}^2.$$

Eventually, leveraging that $\Delta_2^2 \leqslant \Delta_\infty^2 |\mathcal{E}_h|$, we have

$$\zeta^t \leqslant \eta\mu_{\max}(L_h)\left(\|C_h^T Z_h^t\|_{2,-\kappa^t}^2 + (\tau^t)^2\left[\kappa^t(1 - 2\Delta_\infty) + \Delta_\infty^2|\mathcal{E}_h|\right]\right) + 2\|C_h^T Z_h^t\|_{2,\kappa^t}^2.$$

$\square$

**Theorem 2.** *Hence, under Definition 3, for*

$$\eta \leqslant \frac{\mu_{\max}(L_h)^{-1}}{1 + |\mathcal{E}_h|(1 - \Delta_\infty)},$$

*we have*

$$\|Z_h^{t+1}\|^2 \leqslant \|Z_h^t\|^2 - \eta\|C_h^T Z_h^t\|_{2,-\kappa^t}^2.$$

*Proof.* We start from Lemma 4 and use the majoration of the error from Lemma 6:

$$
\begin{aligned}
\|Z_h^{t+1}\|^2 &= \|Z_h^t\|^2 - 2\eta\|C_h^T Z_h^t\|^2 + \eta^2\|L_h Z_h^t - E^t\|^2 + 2\eta\langle Z_h^t - \overline{Z_h^t}, E^t\rangle + 2\langle\overline{Z_h^t}, \overline{Z_h^{t+1}} - \overline{Z_h^t}\rangle \\
&\leqslant \|Z_h^t\| - 2\eta\|C_h^T Z_h^t\|_2^2 + +2\eta\|C_h^T Z_h^t\|_{2,\kappa^t}^2 \\
&\quad + \eta^2\mu_{\max}(L_h)\left(\|C_h^T Z_h^t\|_{2,-\kappa^t}^2 + (\tau^t)^2\left[\kappa^t(1 - 2\Delta_\infty) + \Delta_\infty^2|\mathcal{E}_h|\right]\right) \\
&= \|Z_h^t\| - 2\eta\|C_h^T Z_h^t\|_{2,-\kappa^t}^2 \\
&\quad + \eta^2\mu_{\max}(L_h)\left(\|C_h^T Z_h^t\|_{2,-\kappa^t}^2 + (\tau^t)^2\left[\kappa^t(1 - 2\Delta_\infty) + \Delta_\infty^2|\mathcal{E}_h|\right]\right) \\
&\leqslant \|Z_h^t\| - \eta\left[2 - \eta\mu_{\max}(L_h)\left(1 + \kappa^t(1 - 2\Delta_\infty) + \Delta_\infty^2|\mathcal{E}_h|\right)\right]\|C_h^T Z_h^t\|_{2,-\kappa^t}^2.
\end{aligned}
$$

Where we used that, as $\kappa^t \leqslant \sum_{(i\sim j)\in\mathcal{E}_h} 1_{\|z_i^t - z_j^t\|\geqslant\tau^t} - 1$, the largest term in $\|C_h^T Z_h^t\|_{2,-\kappa^t}^2$ is $(\tau^t)^2$. Hence, for

$$
\eta \leqslant \frac{\mu_{\max}(L_h)^{-1}}{1 + \kappa^t(1 - 2\Delta_\infty) + \Delta_\infty^2|\mathcal{E}_h|},
$$

or, using a simpler (and stricter) condition

$$
\eta \leqslant \frac{\mu_{\max}(L_h)^{-1}}{1 + |\mathcal{E}_h|(1 - \Delta_\infty)},
$$

we have

$$
\|Z_h^{t+1}\|^2 \leqslant \|Z_h^t\|^2 - \eta\|C_h^T Z_h^t\|_{2,-\kappa^t}^2.
$$

$\square$

### B.2.1   On the computation of $\Delta_\infty$

We recall that

$$
\Delta_\infty := \sup_{\substack{U_b\in\mathbb{R}^{\mathcal{E}_b\times d} \\ \|U_b\|_{\infty,2}\leqslant 1}} \|C_h^\dagger C_b U_b\|_{\infty,2}.
$$

**Proposition 4. *(Computing $\Delta_\infty$)*** *Consider a fully connected graph $\mathcal{G}$, we have that*

$$
\Delta_\infty \leqslant \sup_{(i\sim j)\in\mathcal{E}_h} \frac{2n_b}{n_h}\sqrt{d\wedge|\mathcal{E}_b|},
$$

*where $n_b$ is the number of Byzantines nodes and $n_h$ the number of honest nodes.*

*Proof.* In a fully connected graph $L_h = n_h I_h - \mathbf{1}_h\mathbf{1}_h^T$, thus $C_h^\dagger = C_h^T L_h^\dagger = \frac{1}{n_h}C_h^T$. Using this, we compute:

$$
\begin{aligned}
\Delta_\infty &= \sup_{\substack{U_b\in\mathbb{R}^{\mathcal{E}_b\times d} \\ \|U_b\|_{\infty,2}\leqslant 1}} \|C_h^\dagger C_b U_b\|_{\infty,2} \\
&= \frac{1}{n_h}\sup_{\substack{U_b\in\mathbb{R}^{\mathcal{E}_b\times d} \\ \|U_b\|_{\infty,2}\leqslant 1}} \sup_{(i\sim j)\in\mathcal{E}_h} \|[C_h^T C_b U_b]_{(i\sim j),:}\|_2.
\end{aligned}
$$

For $(i \sim j) \in \mathcal{E}_h$,

$$
\begin{aligned}
\sup_{\substack{U_b \in \mathbb{R}^{\mathcal{E}_b \times d} \\ \|U_b\|_{\infty,2} \leqslant 1}} \|[C_h^T C_b U_b]_{(i \sim j),:}\|_2^2 &= \sup_{\substack{U_b \in \mathbb{R}^{\mathcal{E}_b \times d} \\ \|U_b\|_{\infty,2} \leqslant 1}} \sum_{k=1}^{d} \left([C_h^T C_b]_{(i \sim j),:}^T [U_b]_{:,k}\right)^2 \\
&= \sup_{\substack{U_b \in \mathbb{R}^{\mathcal{E}_b \times d} \\ \|U_b\|_{\infty,2} \leqslant 1}} \sum_{k=1}^{d} \left([[C_b]_{i,:} - [C_b]_{j,:}]^T [U_b]_{:,k}\right)^2 \\
&\leqslant \sup_{\substack{U_b \in \mathbb{R}^{\mathcal{E}_b \times d} \\ \|U_b\|_{\infty,2} \leqslant 1}} \sum_{k=1}^{d} \|[[C_b]_{i,:} - [C_b]_{j,:}]\|_1^2 \|[U_b]_{:,k}\|_\infty^2 \\
&\leqslant (N_b(i) + N_b(j))^2 \sup_{\substack{U_b \in \mathbb{R}^{\mathcal{E}_b \times d} \\ \|U_b\|_{\infty,2} \leqslant 1}} \sum_{k=1}^{d} \|[U_b]_{:,k}\|_\infty^2 \\
&\leqslant (N_b(i) + N_b(j))^2 (d \wedge |\mathcal{E}_b|).
\end{aligned}
$$

Hence, we get:

$$
\Delta_\infty \leqslant \sup_{(i \sim j) \in \mathcal{E}_h} \frac{N_b(i) + N_b(j)}{n_h} \sqrt{d \wedge |\mathcal{E}_b|}.
$$

Thus, assuming that, as we are in a fully-connected graph, every honest node is connected to $n_b$ Byzantine node, we get the result. $\qquad\square$

## B.3 A simplified global clipping rule

The following proposition provides a sufficient condition for being a GCR that only requires $\kappa_t$ to be large enough, without any dependence on $X^t$.

**Proposition 5** (Simplified GCR). *If $(\tau^t)_{t \geqslant 0}$ is such that: for all $t \geqslant 0$, either $\tau^t = 0$ or*

$$
\kappa^t \geqslant \Delta_\infty |\mathcal{E}_h| + \eta \frac{|\mathcal{E}_b|^2}{n_h} \sum_{s \leqslant t} \frac{\tau^s}{\tau^t}, \tag{15}
$$

*then $(\tau^s)_{s \geqslant 0}$ satisfies the GCR.*

This result reads as a lower bound on the fraction $\kappa^t/|\mathcal{E}_h|$ of edges that are clipped.

*Proof.* (GCR), i.e Equation (13) writes

$$
\|C_h^T Z_h^t\|_{1,\kappa^t} \geqslant \Delta_\infty \|C_h^T Z_h^t\|_1 + \frac{\eta |\mathcal{E}_b|^2}{n_h} \sum_{s=0}^{t} \tau^s.
$$

We leverage that $\|C_h^T Z_h^t\|_{1,\kappa^t} \geqslant \tau^t \kappa^t$ and that $\|C_h^T Z_h^t\|_{1,-\kappa^t} \leqslant \tau^t(|\mathcal{E}_h| - \kappa^t)$ to write

$$\|C_h^T Z_h^t\|_{1,\kappa^t} \geqslant \Delta_\infty \|C_h^T Z_h^t\|_1 + \frac{\eta|\mathcal{E}_b|^2}{n_h} \sum_{s=0}^t \tau^s$$

$$\iff (1 - \Delta_\infty)\|C_h^T Z_h^t\|_{1,\kappa^t} \geqslant \Delta_\infty \|C_h^T Z_h^t\|_{1,-\kappa^t} + \frac{\eta|\mathcal{E}_b|^2}{n_h} \sum_{s=0}^t \tau^s$$

$$\impliedby (1 - \Delta_\infty)\kappa^t \tau^t \geqslant \Delta_\infty \tau^t(|\mathcal{E}_h| - \kappa^t) + \frac{\eta|\mathcal{E}_b|^2}{n_h} \sum_{s=0}^t \tau^s$$

$$\iff \kappa^t \tau^t \geqslant \Delta_\infty \tau^t |\mathcal{E}_h| + \frac{\eta|\mathcal{E}_b|^2}{n_h} \sum_{s=0}^t \tau^s$$

$$\iff \kappa^t \geqslant \Delta_\infty |\mathcal{E}_h| + \frac{\eta|\mathcal{E}_b|^2}{n_h} \sum_{s=0}^t \frac{\tau^s}{\tau^t}.$$

$\square$

