# OpenReview forum: "Byzantine-Robust Gossip: Insights from a Dual Approach"
_TMLR — Accepted by TMLR_

### Review · Reviewer_YXxX · 2025-07-24

**Summary Of Contributions:**

This paper introduces EdgeClippedGossip, a new Byzantine-robust algorithm for decentralized learning that uses a dual optimization framework to clip influence updates on graph edges. It provides convergence guarantees for this method in the average consensus setting, where it recovers existing algorithms. It also proposes a Global Clipping Rule (GCR) as a theoretical proof-of-concept, demonstrating a trade-off where GCR reduces bias but doesn't guarantee full consensus. While GCR has practical limitations due to its global coordination requirement, the paper re-interprets other robust algorithms within its dual framework and highlights the challenges of extending this approach to more general decentralized optimization problems.

**Audience:**

Yes

**Claims And Evidence:**

Yes

**Requested Changes:**

- In the 2nd paragraph of Introduction, there is a typo: "We denote $V_b$ as Byzantine nodes...".
- Section 7, Statement of Broader Impact, is empty.

**Strengths And Weaknesses:**

**Strengths**
1. The paper introduces a novel perspective by leveraging the dual approach for Byzantine robustness in decentralized optimization. It connects with existing robust communication algorithms, providing a unifying view.
2. The paper provides theoretical convergence guarantees for the proposed EdgeClippedGossip in the average consensus subcase under the Global Clipping Rule (GCR).
3. The paper demonstrates the trade-off between bias and variance, and shows empirical results supporting this claim.


**Weaknesses**
1. The paper is limited in proof of concept for the Global Clipping Rule, but doesn't provide a practical decentralized algorithm.
2. The experimental setup is unclear. The whole paper is based on decentralized settings, but it compares with FOE and ALIE which are federated settings with a central server. Why is it necessary to compare with these two baselines rather than other decentralized methods?
3. Empirical results are insufficient. Could you provide empirical results on MNIST and CIFAR-10? Although implementing GlobalClipping may be challenging, comparing it with LocalClipping and LocalTrimming on more realistic datasets is essential.

---

> ### Author Response · Authors · 2025-10-17
>
> We thank the reviewer for their attentive reading and their comments, which we discuss below.
>
> # Practicality
>
> We do agree with the reviewer that the practicality of the proposed method has some limits.
>
> However, we would like to point out that the raised concern can be circumvented by the two following points:
> 1. As answered to Reviewer GPXg, since the pair-wise differences $\|x_i-x_j\|$ are one-dimentional vectors, broadcasting securely this information to all the other nodes can be performed with a reasonably small communication overhead. For instance, Signed Echo Broadcast (SEB) [1] allows it in a Byzantine-resilient manner.
> 2. As we point out in the paragraph *Over-Clipping* of Section 3.1, if all pair-wise differences are known, conservative strategies can be used to derive a threshold that respects the Global Clipping rule.
>
>
>
> # Attack baseline
>
> Our experimental methodology relies on testing both decentralized attacks (Spectral Heterogeneity and Dissensus) as well as attacks that were first designed in the federated setting (FOE and ALIE). Importantly, even if those two previous attacks were first designed in the federated setting, multiple works used them to test their decentralized algorithms, with slight modifications to adapt them to the setting (see e.g. [2], [3] [4]). Furthermore, the two decentralized attacks tested in our work are the only ones specifically designed for the decentralized setting we are aware of. If the reviewer knows any other relevant attack, we would be delighted to implement and test them.
>
> NB: attacks such as gradient flipping and label flipping only make sense when local models are actually optimized, which is not the case for plain gossip communication.
>
> # Empirical Evaluation
>
> We believe that the key takeaway of our works rely on the introduction of an original framework, that allows both to understand existing algorithms, and that originally targets slightly different metrics - since it does not try to enforce consensus. Moreover, investigating the dual approach was a natural path towards efficient resilient decentralized optimization algorithms, and we believe that our discussion on the hardness of the approach is valuable to the community.
>
> Consequently, our experiments are mostly intended to illustrate the fundamental differences between our method and sota approaches. We do not claim that the dual framework with Global Clipping is currently a competitive algorithm for optimizing more generic loss functions than the sum of squared distance: the Global Clipping rule is derived for this specific loss, and would not work for a more generic one. More importantly, if the dual approach has historically been used extensively for designing optimal decentralized algorithms, the dual-only algorithms that rely on Fenchel gradients are not practical in general, and served as stepping stones for developing primal versions. Given these considerations, we believe that adding further experiments of the Global Clipping method with more generic loss functions would not serve the point we want to make with this paper.
>
>
> __
>
> [1] Cachin, C., Guerraoui, R., & Rodrigues, L. (2011). Introduction to reliable and secure distributed programming. Springer Science & Business Media
>
> [2] He, L., Karimireddy, S. P., & Jaggi, M. (2022). Byzantine-robust decentralized learning via clippedgossip. arXiv preprint arXiv:2202.01545.
>
> [3] Farhadkhani, S., Guerraoui, R., Gupta, N., Hoang, L. N., Pinot, R., & Stephan, J. (2023, July). Robust collaborative learning with linear gradient overhead. In International Conference on Machine Learning (pp. 9761-9813). PMLR.
>
> [4] Gaucher, R., Dieuleveut, A., & Hendrikx, H. Unified Breakdown Analysis for Byzantine Robust Gossip. In Forty-second International Conference on Machine Learning.

---

### Review · Reviewer_unrv · 2025-08-19

**Summary Of Contributions:**

This paper conducts a theoretical study of Byzantine-robust distributed learning in the decentralized setting. The authors propose EdgeClippedGossip which is robust against Byzantine nodes that is designed based on clipped dual gradient descent. Convergence guarantee is provided for the proposed algorithm for global clipping. The paper also uses the proposed framework to re-interpret previous methods based on local clipping.

**Audience:**

Yes

**Broader Impact Concerns:**

The Broader Impact section is left empty. I think the paper is indeed more theoretical in nature and hence may not have immediate negative impacts.

**Claims And Evidence:**

Yes

**Requested Changes:**

Please refer to the Weaknesses above. I think both the theoretical and empirical aspects of the paper could benefit from additional contributions.

**Strengths And Weaknesses:**

Strengths:
- The paper uses the dual optimization framework to develop a principled method for Byzantine-robust decentralized optimization. The proposed method is theoretically grounded.
- The paper also uses the dual framework to re-interpret and discuss existing local clipping algorithms. Based on this, the paper reveals an interesting trade-off: the local clipping-based methods introduce additional biases but enjoys guaranteed asymptotic consensus; on the other hand, the global clipping-based method is free from such biases yet does not guarantee consensus.
- The authors are transparent about the limitations of the paper, for example, the theoretical result for the global clipping method does not provide convergence rates.

Weaknesses:
- If I understand correctly, the proposed global clipping strategy is in fact an oracle strategy, in that it requires knowledge of the "sum of node-wise differences on honest neighbors only". This makes this strategy impractical in practice, and hence may limit the practical contribution of the paper.
- As the authors have acknowledged, an important limitation of the theoretical results in this paper is that it does not offer a concrete convergence rate. This is likely to limit the theoretical contributions of the paper.
- It is unclear to me how the empirical results in Fig. 2 demonstrate the empirical advantage of the proposed methods. The results suggest that previous methods based on local clipping consistently achieve smaller final errors, and the proposed global clipping method is likely overly conservative in practice.

---

> ### Author Response · Authors · 2025-10-17
>
> We thank the reviewer for their attentive reading.
>
> # Practicality
>
> As the reviewer correctly understood, finding the tightest clipping thresholds in the Global Clipping condition assumes to have the knowledge of the node-wise differences on honest neighbors only.
>
> However, as we point out in our paper (bottom Section 3.1), an overly conservative but non-oracle clipping threshold can be derived as well, which rely on the pair-wise differences of all parameters, and not the honest ones only.
>
> Furthermore, as we point out in the answer to Reviewer GPXg, having access to the node-wise differences $\||x_i^t-x_j^t|\|$ is not an un-reasonable assumption, as those are one-dimensional quantities, and as such significantly cheaper to communicate. Hence, reling for instance on Signed Echo Broadcast (SEB) [1] to share those information with all the other nodes could be a reasonable solution, which can maintain a low communication cost.
>
>
>
>
>
> # Convergence rate
>
>
> We agree that showing a proper convergence rate for a dual approach in a Byzantine setting would be highly valuable for the community. However, as argued in Section 4.2, this is highly non-trivial, since the addition of Byzantine agents breaks the structure of the dual problem.
>
>
> # Empirical evaluations
>
> As the reviewer points out, the Global Clipping strategy consistently achieves lower accuracy than state of the art local aggregation strategy. However, we believe our approach still offers an alternative point of view in the design of Byzantine resilient communication algorithms, since it’s the first one that tries to directly optimize for the optimization error, without requiring to reach consensus. That not reaching consensus could actually be interesting for Byzantine resilient algorithms does not appear in the literature (up to our knowledge), thus our paper provides an alternative view in the design of such algorithms.
>
>
>
> __
>
> [1] Cachin, C., Guerraoui, R., & Rodrigues, L. (2011). Introduction to reliable and secure distributed programming. Springer Science & Business Media

---

### Review · Reviewer_GPXg · 2025-10-09

**Summary Of Contributions:**

This paper presents a study on Byzantine-robust decentralized optimization, specifically within the gossip communication framework. Its primary contributions includes:

(1)The paper leverages the dual approach to decentralized optimization to derive EdgeClippedGossip, a new algorithm that achieves robustness by applying a clipping operator directly to the updates of the Lagrange multipliers (the "influence" between nodes) on the edges of the communication graph.

(2)The authors provide a rigorous theoretical analysis of EdgeClippedGossip in the Average Consensus Problem (ACP). They introduce a Global Clipping Rule (GCR) and, under this rule, prove convergence guarantees (Theorem 1). This analysis formally characterizes the trade-off between reducing variance among honest nodes and controlling the bias introduced by Byzantine nodes.

(3) The paper demonstrates that several existing state-of-the-art robust gossip algorithms (e.g., He et al., 2023; Gaucher et al., 2025) can be recovered as specific instances of their proposed dual framework with asymmetric clipping. Furthermore, it offers a insightful discussion on the fundamental challenges—such as the loss of the invariant property and the lack of a Lyapunov function—that arise when extending the dual approach to Byzantine settings beyond simple averaging.

**Audience:**

Yes

**Claims And Evidence:**

Yes

**Requested Changes:**

The related work section and the broader discussion could be improved by a more direct comparison with other robust aggregation strategies. For instance, while the clipping operation is central to this work, it would be insightful to relate it to other robust loss functions or gradient manipulations.

For example, the work of Zhao, Puning, Fei Yu, and Zhiguo Wan (2024) on Huber loss minimization for Byzantine-robust federated learning offers an alternative, well-motivated statistical approach to outlier rejection. A comparison of the underlying philosophy (loss geometry vs. dual influence control) would be valuable.

Similarly, the gradient normalization technique proposed by Zuo, Shiyuan, et al. (2024) for handling non-IID data in federated learning addresses the challenge of biased updates, which is related to the "bias" discussed in this paper. Drawing connections to such methods could broaden the paper's relevance.

The very recent work by Fang, Minghong, et al. (2024) on "Byzantine-robust decentralized federated learning" is a direct contemporary; a more detailed comparison to their approach and findings would better situate this paper within the current literature.

[1]Zhao, Puning, Fei Yu, and Zhiguo Wan. "A huber loss minimization approach to byzantine robust federated learning." Proceedings of the AAAI Conference on Artificial Intelligence. Vol. 38. No. 19. 2024.

[2]Zuo, Shiyuan, et al. "Byzantine-resilient federated learning employing normalized gradients on non-iid datasets." arXiv preprint arXiv:2408.09539 (2024).

[3]Fang, Minghong, et al. "Byzantine-robust decentralized federated learning." Proceedings of the 2024 on ACM SIGSAC Conference on Computer and Communications Security. 2024.

**Strengths And Weaknesses:**

The paper has several strengths.

Firstly, the application of the dual formulation to Byzantine robustness is a significant and novel conceptual contribution. It provides a principled and unifying framework for understanding and designing robust decentralized algorithms, moving beyond ad-hoc aggregation rules.

The second factor is theoretical analysis. The convergence analysis for the ACP case is thorough. The derivation of the GCR is a key strength, as it formally justifies when and why communication should be attenuated to prevent Byzantine nodes from causing harm, a concept often implicit but not formally captured in prior work.

In general, I would like to recommend this paper for acceptance. However, there are also several weaknesses.

(1) Limited Practicality of the GCR. The primary weakness of the proposed GCR is its reliance on global coordination and knowledge (e.g., of $\Delta_{\infty}$, $|C_h^T X_h^t|_{1,2;\kappa^t}$). The authors correctly note this is a "proof of concept," which limits the immediate applicability of the main theoretical result in truly decentralized systems. A discussion on how to approximate these global quantities in a decentralized manner would significantly strengthen the paper's impact.

(2) Narrow Problem Scope. The strong theoretical guarantees are currently confined to the Average Consensus Problem. While Section 4 provides an excellent discussion of the difficulties in generalizing the approach, the paper would be significantly more impactful if it included even a preliminary extension or empirical validation on a more general distributed optimization problem (e.g., distributed logistic regression).

---

> ### Author Response · Authors · 2025-10-17
>
> We thank the reviewer for their careful reading and their questions, which we answer below.
>
> # Decentralized approximation of global quantities
>
> We agree with the reviewer that discussing this issue is a valuable addition to our work
>
> ## Estimating $\Delta_{\infty}$
>
> We agree that it is hard in practice to exactly compute $\Delta_{\infty}$, especially since the relative position of Byzantine nodes is unknown. We would still like to point out that most Byzantine-resilient algorithms rely on an assumption of the amount of Byzantine contamination in the system. In this sense, $\Delta_{\infty}$ is just an abstract one that combines the topology of the graph with the position of Byzantine nodes. Yet, we agree that $\Delta_{\infty}$ is a very abstract quantity, which can be hard in practice to derive from more standard assumptions.
>
>
> One way to use simpler quantities such as a number of Byzantine neighbors per node instead of $\Delta_{\infty}$, could be to estimate $\Delta_{\infty}$ by relying on approximation of the matrices $C_h^{\dagger}$ and $C_b$.
>
>
> We believe that, if the global communication graph is known, one reasonably good estimation procedure is the following:
> 1. First, we compute the pseudo-inverse $C_h^{\dagger}$ by assuming that all nodes are honest, which is a good approximation if there are only a few Byzantine nodes with random positions, and the graph is sufficiently connected.
> 2. One can then build $C_b$ by assuming, for instance, that each honest node has at most f (given) Byzantine nodes. Which is a standard hyper-parameter used in the literature.
> 3. Now that both matrices have been estimated, we can explicitly compute $\Delta_{\infty}$ if d=1 (since it’s the largest l1 norm among rows of $C_h^{\dagger}C_b$), or estimate it using optimization routines for d>1.
>
>
> ## Estimating the $\|x_i - x_j\|$.
>
>
> To be computed the Global Clipping threshold requires to have access to all $(\|x_i^t - x_j^t\|: i,j \in E)$. Since those are one dimensional quantities, broadcasting robustly them to all the other nodes is significantly cheaper (by a factor d) than directly broadcasting robustly the parameter themselves $(x_i)_i$, and it is therefore affordable to do so (even if communicating the models $x_i$ to only a few other nodes is targeted to minimize the communication costs).
>
> One way to perform this broadcast in a secure manner could be to build on the Signed Echo Broadcast (SEB) protocol [4], which requires that each node communicates O(n) data.
>
> [4] Cachin, C., Guerraoui, R., & Rodrigues, L. (2011). Introduction to reliable and secure distributed programming. Springer Science & Business Media
>
>
>
>
> # Implementation of more general distributed optimization problem
>
> We believe that the key takeaway of our work relies on the introduction of an original framework, that allows both to understand existing algorithms, and suggest to target slightly different metrics than SOTA algorithms - since it does not try to enforce consensus. Moreover, investigating the dual approach was a natural path towards efficient resilient decentralized optimization algorithms, and we believe that our discussion on the hardness of the approach is valuable to the community.
>
>
> Consequently, our experiments are mostly intended to illustrate the fundamental differences between our method and sota approaches. We do not claim that the dual framework with Global Clipping is currently a competitive algorithm for optimizing more generic loss functions than the sum of squared distance: the Global Clipping rule is derived for this specific loss, and would not work for a more generic one. More importantly, if the dual approach has historically been used extensively for designing optimal decentralized algorithms, the dual-only algorithms that rely on Fenchel gradients are not practical in general, and served as stepping stones for developing primal versions. Given these considerations, we believe that adding further experiments of the Global Clipping method with more generic loss functions would not serve the point we want to make with this paper.

---

> ### Author Response · Authors · 2025-10-17
>
> # Related works
>
>
> We thank the reviewer for pointing out those work, which we were not aware of. We point out similarities with respect to our work below.
>
> ## Link with Huber loss
>
>  There is, indeed, a deep connection between the control of the dual variables and Huber-loss minimization approaches. Indeed, one step of centered clipping similar to
> $$
> x_i^{t+1} = x_i^t - \eta \sum_{j} clip(x_i^t - x_j^t; c)
> $$
> is equivalent to performing one GD step on the objective minimized by the Huber estimator, with initial parameter $x_i^t$, namely
> $$
> \min_{x} \phi(x) := \sum_{j} \psi_c(x-x_i)
> $$
> where $\psi_c$ denotes the Huber loss defined by
>
> \\begin{equation}
> \psi_c(x)=
> \\begin{cases}
> 1/2 \|x\|^2 \quad if \quad |x|<c\\
> c( |x| - 1/2 c)  if \quad |x| \ge c
> \\end{cases}
> \\end{equation}
>
> From a dual perspective, clipping the update of the Lagrangian multipliers is thus equivalent to performing a gradient descent step on the Huber Estimation Loss $\phi$ with a good initialization, instead of the mean squared distance.
>
> ## Link with gradient normalization aggregation
>
> We thank the reviewer for pointing out this paper. And we would like to emphasize small but important differences between the “bias” raised in Zua, Shiyuan et al (2024) and the “bias” we are referring to in our work.
>
>
> The bias raised by Zua, Shiyuan et al (2024) corresponds to the minimal achievable error of any optimization algorithm, for instance federated ones (ie server-based), when loss functions are heterogeneous and Byzantine workers are involved. See for instance Karimireddy et al. (2020), or Allouha et al. (2023).
>
> Differently, the bias we are referring to corresponds to one which is specific to decentralized approaches and to the communication used in this setting, and corresponds to the distance of the actual average of parameter $\overline{X}^t$ to the initial average of parameter $\overline{X}^\*$. We called this error on the average “bias” in order to better understand the error dynamic as a bias-variance decomposition. This bias $\overline{X}^t - \overline{X}^\*$ can be equal to 0, for instance if nodes do not communicate, which is not the case for the optimization error.
>
> Note that this bias-variance decomposition only makes sense for a decentralized system, since it’s harder (and arguably not necessary) to enforce consensus on the node’s parameters across iteration.
>
>
> Indeed, in the federated setting, approaches such as Zua, Shiyuan et al (2024) enforce that nodes compute their gradient on the same parameter, sent at each iteration by the central server. This reduces the bias $\overline{X}^t - \overline{X}^*$ to the whole optimization error.
>
>
>
> ## Byzantine-robust decentralized FL
>
> This work focuses on a very different problem: they assume that nodes have iid data, and they mostly try to develop algorithms that do not use the upper bound in the number of Byzantine nodes as a parameter. Furthermore, they make very light assumptions on the connectivity of the network. As a consequence, they do not (and, to our belief, cannot) show much improvement of the convergence rate due to the communication. Thus, their main challenge is to show that communicating with Byzantine nodes is not too harmful to the process, while still maintaining convergence results. For instance, their Theorem 1 controls the loss induced by communication with potentially corrupted nodes, compared to the one induced by local optimization steps only.
>
>
> On the contrary, we try to tackle the heterogeneous setting, in which nodes have different loss functions. This implies that a Byzantine-resilient communication method is necessary to get closer to the optimum.
>
> __
>
>
> Karimireddy, S. P., He, L., & Jaggi, M. (2020). Byzantine-robust learning on heterogeneous datasets via bucketing. arXiv preprint arXiv:2006.0936
>
> Allouah, Y., Guerraoui, R., Gupta, N., Pinot, R., & Rizk, G. (2023). Robust distributed learning: Tight error bounds and breakdown point under data heterogeneity. Advances in Neural Information Processing Systems, 36, 45744-45776.

---

### Decision · Action_Editor_mPTu · 2025-12-31

**Recommendation:** Accept as is

**Audience:**

Yes

**Audience Explanation:**

The problem of robustness to Byzantine nodes in a decentralized setting is relevant and interesting to TMLR community.

**Claims And Evidence:**

Yes

**Claims Explanation:**

The authors have studied robustness to Byzantine nodes in a decentralized setting and proposed a dual optimization framework that clips updates on graph edges. They have established convergence guarantees in the average consensus setting recovering existing robust gossip methods. They also proposed the Global Clipping Rule as a theoretical proof-of-concept and showed that the Global Clipping Rule reduces bias while it does not guarantee full consensus.

The reviewers have highlighted several strengths in this paper. For example, Reviewer YXxX noted “a novel perspective by leveraging the dual approach for Byzantine robustness in decentralized optimization”. Reviewer unrv wrote “The proposed method is theoretically grounded.” Reviewer GPXg noted “the application of the dual formulation to Byzantine robustness is a significant and novel conceptual contribution. It provides a principled and unifying framework for understanding and designing robust decentralized algorithms, moving beyond ad-hoc aggregation rules” and “The convergence analysis for the ACP case is thorough. The derivation of the GCR is a key strength”.

After revisions and discussion with reviewers, some reviewers still have some concerns including “lack of a convergence rate” and “the proposed method yields higher error rates than existing local aggregation methods”.

Overall, I think providing more realistic experiments and extension to more general problems in terms of loss function would be helpful as future work. Given novelty perspective of using dual approach for Byzantine resilience in decentralized setting, I would recommend the paper to be accepted as is.